# Generalization as Dynamical Robustness–
# The Role of Riemannian Contraction in Supervised Learning

**Leo Kozachkov**                                                                *leokoz8@mit.edu*
*Department of Brain and Cognitive Sciences*
*K. Lisa Yang Integrative Computational Neuroscience (ICoN) Center*
*Massachusetts Institute of Technology*

**Patrick M. Wensing**                                                           *pwensing@nd.edu*
*Department of Aerospace and Mechanical Engineering*
*University of Notre Dame*

**Jean-Jacques Slotine**                                                         *jjs@mit.edu*
*Department of Mechanical Engineering*
*Department of Brain and Cognitive Sciences*
*Massachusetts Institute of Technology*

**Reviewed on OpenReview:** *https://openreview.net/forum?id=Sb6p5mcefw*

## Abstract

A key property of successful learning algorithms is generalization. In classical supervised learning, generalization can be achieved by ensuring that the empirical error converges to the expected error as the number of training samples goes to infinity. Within this classical setting, we analyze the generalization properties of iterative optimizers such as stochastic gradient descent and natural gradient flow through the lens of dynamical systems and control theory. Specifically, we use contraction analysis to show that generalization and dynamical *robustness* are intimately related through the notion of algorithmic stability.

In particular, we prove that Riemannian contraction in a supervised learning setting implies generalization. We show that if a learning algorithm is contracting in some Riemannian metric with rate $\lambda > 0$, it is uniformly algorithmically stable with rate $\mathcal{O}(1/\lambda n)$, where $n$ is the number of examples in the training set. The results hold for stochastic and deterministic optimization, in both continuous and discrete-time, for convex and non-convex loss surfaces.

The associated generalization bounds reduce to well-known results in the particular case of gradient descent over convex or strongly convex loss surfaces. They can be shown to be optimal in certain linear settings, such as kernel ridge regression under gradient flow. Finally, we demonstrate that the well-known Polyak-Lojasiewicz condition is intimately related to the contraction of a model's *outputs* as they evolve under gradient descent. This correspondence allows us to derive uniform algorithmic stability bounds for nonlinear function classes such as wide neural networks.

## 1 Introduction

Since the seminal work of Bousquet & Elisseeff (2002), the concept of *algorithmic stability* has been used to analyze the generalization properties of learning algorithms (Mukherjee et al., 2006; Shalev-Shwartz et al., 2009; 2010; Hardt et al., 2016). Roughly speaking, algorithmic stability refers to the notion that small changes to the training set will lead to small changes in the output of the learning process.

In this work, we focus on iterative optimizers within a supervised learning setting, where we are given access to a number of labelled training points drawn from some underlying common distribution, as well as a loss

function which quantifies performance. Within this setting, we show that algorithmic stability is intimately related to notions of *robustness* from the dynamical systems and control literature. In particular, we make a connection between algorithmic stability and contractive stability (Lohmiller & Slotine, 1998). Loosely, a dynamical system is contracting if it forgets its initial conditions exponentially quickly.

Contraction analysis has found wide application in nonlinear control theory (Manchester & Slotine, 2017), robotics (Chung & Slotine, 2009), and synchronization (Pham & Slotine, 2007). However, it has only recently been applied to machine learning (Boffi et al., 2020; Revay & Manchester, 2020; Jafarpour et al., 2021; Burghi et al., 2022). We show that if an optimizer is *contracting* (Lohmiller & Slotine, 1998) in some Riemannian metric (in a precise sense defined below) then it is algorithmically stable. Due to the generality of contraction analysis and the flexibility afforded us by the choice of metric, our theory applies to wide variety of common optimizers–for example gradient flows and stochastic minibatch gradient descent–operating over both convex and non-convex loss surfaces (see Figures 1, 2, and 3).

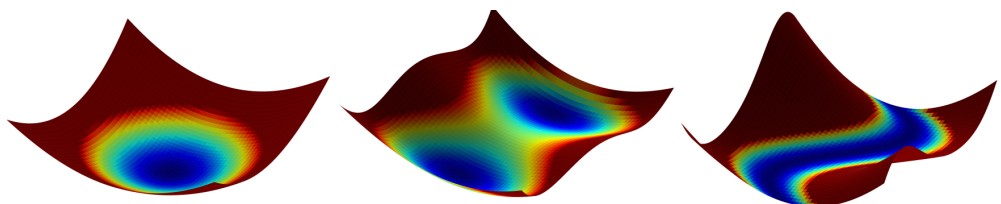

Figure 1: Example loss surfaces for which our results apply. Left panel: strongly convex and convex loss surfaces (sections 3.1 and 4.2.1). Middle panel: isolated local minima surrounded by basins of contraction (see Theorem 7). Right panel: valley of path-connected global minima (see section 4.2).

## 1.1 Related Work

A key early result in analyzing generalization in iterative optimization came from (Hardt et al., 2016), which established algorithmic stability for stochastic gradient methods. Later (Mou et al., 2018) proved similar results for stochastic gradient Langevin dynamics. Shortly thereafter (Charles & Papailiopoulos, 2018) showed that for loss functions satisfying certain geometrical constraints (e.g., the Polyak-Łojasiewicz inequality (Polyak, 1963)), any optimizer that converges to a global minimum is also algorithmically stable. Since then, several follow-up works have analyzed the algorithmic stability of accelerated gradient methods, and the tradeoffs between optimization accuracy and algorithmic stability (Chen et al., 2018; Ho et al., 2020; Attia & Koren, 2021). The present work is similar in spirit to Charles & Papailiopoulos (2018), in the sense that we use an assumed stability property (in our case, contraction of optimizer trajectories) to derive generalization bounds for a wide class of optimizers. The following section introduces our supervised learning setting, which is the same setting as in Hardt et al. (2016), and provides necessary background on algorithmic stability.

## 1.2 Algorithmic Stability Background

We consider a generic supervised learning setting where we have access to $n$ labelled examples, assumed to be drawn i.i.d from an unknown distribution $\mathcal{D}$ (Vapnik, 1999). We collect these examples into a training set $S = (z_1, \ldots, z_n)$. The *population risk* with respect to a loss function $\ell$ is defined as

$$R[\boldsymbol{\theta}] = \mathbb{E}_{z \sim D} \, \ell(\boldsymbol{\theta}, z)$$

where $\boldsymbol{\theta} \in \mathbb{R}^m$ describes a model. We assume that we do not know the population risk, so we use the *empirical risk* as a proxy

$$R_S[\boldsymbol{\theta}] = \frac{1}{n} \sum_{i=1}^{n} \ell(\boldsymbol{\theta}, z_i)$$

The difference between the population and empirical risk is denoted as the *generalization error* of model $\boldsymbol{\theta}$

$$\Delta^{gen}(\boldsymbol{\theta}) \equiv R[\boldsymbol{\theta}] - R_S[\boldsymbol{\theta}]$$

We now define the stability of an algorithm, and relate it to this generalization error. Consider an algorithm $\mathcal{A}$ that takes in $S$ and outputs a model (e.g., a parameter vector $\boldsymbol{\theta}$).

**Definition 1** (Uniform Algorithmic Stability). An algorithm $\mathcal{A}$ is $\epsilon$-uniformly stable if for all data sets $S, S'$ such that $S$ and $S'$ differ in at most one example, we have

$$\sup_z \mathbb{E}_{\mathcal{A}}[\ell(\mathcal{A}(S), z) - \ell(\mathcal{A}(S'), z)] \leq \epsilon \tag{1}$$

where the expectation is taken over the randomness of $\mathcal{A}$, if any.

A key result in learning theory states that uniform stability leads to generalization in expectation (Bousquet & Elisseeff, 2002; Shalev-Shwartz et al., 2010; Hardt et al., 2016). In particular, we use Theorem 2.2 of Hardt et al. (2016).

**Theorem 1.** *Let $\mathcal{A}$ be $\epsilon$-uniform stable and let $\mathbb{E}_{S,\mathcal{A}}$ denote an expectation taken over the samples $S$ and the randomness of $\mathcal{A}$. Then, $|\mathbb{E}_{S,\mathcal{A}}\left[\Delta^{gen}(\mathcal{A}(S))\right]| \leq \epsilon$.*

If the output of $\mathcal{A}$ is some parameter vector $\boldsymbol{\theta}$ and we assume that our loss function is $L$-Lipshitz for every example $z_i$ with respect to some norm $||\cdot||$, then the difference between two trajectories of an optimizer trained on set $S$ and $S'$ can be used to bound the generalization error, because

$$\mathbb{E}_{\mathcal{A}}[|\ell(\boldsymbol{\theta}_S, z) - \ell(\boldsymbol{\theta}_{S'}, z)|] \leq L\,\mathbb{E}_{\mathcal{A}}||\boldsymbol{\theta}_S - \boldsymbol{\theta}_{S'}|| \tag{2}$$

Rather than only considering the Euclidean distance $||\boldsymbol{\theta}_S - \boldsymbol{\theta}_{S'}||$, in this paper we consider the *geodesic distance* $d_{\mathcal{M}}(\boldsymbol{\theta}_S, \boldsymbol{\theta}_{S'})$ computed on a Riemannian manifold $\mathcal{M} = (\mathbb{R}^m, \mathbf{M})$ (Figure 2). Here $\mathbf{M}(\boldsymbol{\theta}, t) \in \mathbb{R}^{m \times m}$ is the positive definite metric associated to $\mathcal{M}$. There are many optimization settings for which the geodesic distance between two points – as opposed to the Euclidean norm – is the more natural distance measure to consider (Amari, 1998; Wensing & Slotine, 2020). The main takeaway of this paper is that *Riemannian contraction implies generalization in supervised learning.* The details about this generalization (e.g., its dependence on the number of samples $n$ and the training time $T$) depend on the dynamical equations of the optimizer, as well as the geometry of the loss landscape, as we will see. We now provide background on nonlinear contraction analysis before stating our results.

## 1.3 Nonlinear Contraction Theory Background

Consider a state vector $\boldsymbol{\theta} \in \mathbb{R}^m$, evolving according to the continuous-time dynamics

$$\dot{\boldsymbol{\theta}} = \mathbf{f}(\boldsymbol{\theta}, t) \tag{3}$$

where it is assumed that all quantities are real and smooth, so any required derivative or partial derivative exists and is continuous. Then we have the following definition

**Definition 2** (Contracting Dynamical System). Denote the Jacobian of (3) by $\mathbf{J} \equiv \frac{\partial \mathbf{f}}{\partial \boldsymbol{\theta}}(\boldsymbol{\theta}, t)$. If there exists a symmetric positive-definite metric $\mathbf{M}(\boldsymbol{\theta}, t) : \mathbb{R}^m \times \mathbb{R} \to \mathbb{R}^{m \times m}$ and a scalar $\lambda > 0$ such that the following *differential Lyapunov matrix equation* is uniformly satisfied in space and time

$$\dot{\mathbf{M}} + \mathbf{M}\mathbf{J} + \mathbf{J}^T\mathbf{M} \leq -2\lambda\mathbf{M} \tag{4}$$

then the geodesic distance defined with respect to $\mathbf{M}$ between any two trajectories of (3) converges to zero exponentially, with rate $\lambda$, and (3) is said to be *contracting*. Discrete-time contraction can be defined similarly (Lohmiller & Slotine, 1998). One may consider more generally in (4) a (uniformly positive definite) instantaneous contraction rare $\lambda(t)$.

### 1.3.1 Robustness of Contracting Systems

Contracting systems are robust to disturbances, in the following sense. Assume that (3) is contracting in metric $\mathbf{M} = \mathbf{T}(\boldsymbol{\theta}, t)^T\mathbf{T}(\boldsymbol{\theta}, t)$ with rate $\lambda$. Now consider the same dynamics as (3), perturbed with some disturbance

$$\dot{\boldsymbol{\theta}}_p = \mathbf{f}(\boldsymbol{\theta}_p, t) + \mathbf{d}(\boldsymbol{\theta}_p, t) \tag{5}$$

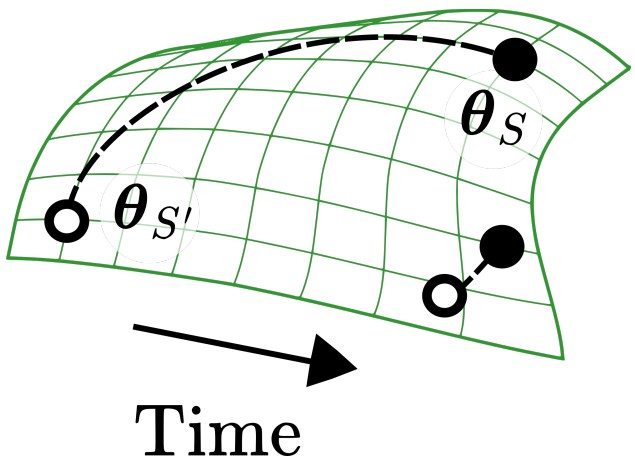

Figure 2: The geodesic distance between optimizer trajectories $\boldsymbol{\theta}_S$ and $\boldsymbol{\theta}_{S'}$. If the optimizer is contracting with rate $\lambda$, this distance, denoted by the dashed line in the figure and $d_{\mathcal{M}}(\boldsymbol{\theta}_S, \boldsymbol{\theta}_{S'})$ in the text, shrinks until the two trajectories are within a ball of radius $\mathcal{O}(\frac{1}{n\lambda})$.

The geodesic distance $d_{\mathcal{M}}(\boldsymbol{\theta}, \boldsymbol{\theta}_p)$ satisfies the *differential inequality*

$$\frac{\mathrm{d}}{\mathrm{d}t}d_{\mathcal{M}}(\boldsymbol{\theta}, \boldsymbol{\theta}_p) + \lambda d_{\mathcal{M}}(\boldsymbol{\theta}, \boldsymbol{\theta}_p) \leq ||\mathbf{T}(\boldsymbol{\theta}, t)\mathbf{d}(\boldsymbol{\theta}_p, t)|| \tag{6}$$

Assuming there exists a finite constant $D$ such that $||\mathbf{d}(\boldsymbol{\theta}_p, t)|| \leq D$ uniformly, (6) implies

$$R(t) \leq \chi R(0)e^{-\lambda t} + \frac{D\chi}{\lambda} \tag{7}$$

where $R(t) \equiv ||\boldsymbol{\theta}(t) - \boldsymbol{\theta}_p(t)||$ and $\chi$ denotes an upper-bound on the condition number of $\mathbf{T}$. Likewise for the discrete-time dynamics contracting in some metric with rate $0 < \mu < 1$

$$\boldsymbol{\theta}_{t+1} = \mathbf{f}(\boldsymbol{\theta}_t, t)$$

the analogous result is

$$R(t) \leq \chi R(0)\mu^t + \frac{D\chi}{(1-\mu)} \tag{8}$$

For proofs of these statements we refer the reader to Lohmiller & Slotine (1998) (section 3.7, vii) as well as Del Vecchio & Slotine (2012) and Proposition 1 in the appendix of Zhang et al. (2021).

If we interpret (3) as an algorithm, then the only source of indeterminacy in this algorithm is the initial condition $\boldsymbol{\theta}(0)$. Therefore if (3) is always initialized within a ball of radius $C/2$ of some reference point, then (7) may be stated in expectation

$$\mathbb{E}_{\mathcal{A}}[R(t)] \leq \mathbb{E}_{\mathcal{A}}[\chi R(0)e^{-\lambda t} + \frac{D\chi}{\lambda}] \leq \chi C e^{-\lambda t} + \frac{D\chi}{\lambda} \tag{9}$$

where we have used the linearity of the expectation value operator, as well as the assumption $\mathbb{E}_{\mathcal{A}}[R(0)] \leq C$.

Let us also state a new robustness result using instantaneous contraction rates, which will be useful later.

**Theorem 2.** *If the instantaneous contraction rate $\lambda(t) > 0$ of a system tends towards a finite limit $\lambda_\infty$, with $\dot{\lambda} \to 0$, then $R(t)$ tends to the bound $0 \leq R(t) \leq \frac{\chi D}{\lambda_\infty}$ .*

*Proof.* By analogy with (Slotine & Coetsee, 1986), define $R_\Delta = R - \phi\,\mathrm{sat}(R/\phi)$ where $\phi \equiv \frac{\chi D}{\lambda(t)}$. One has

$$\frac{d}{dt}R_\Delta^2 + 2\lambda(t)R_\Delta^2 \leq 2R_\Delta\dot{\phi}$$

As $t \to +\infty$, the right-hand side of the above inequality tends to zero, which in turn implies (from the Bellman-Gronwall lemma) that $R_\Delta$ tends to zero, i.e., that $R$ tends to the bound $R \leq \phi_\infty = \frac{\chi D}{\lambda_\infty}$. $\qquad\square$

### 1.3.2 Geodesics and Bounded Distortions

To ensure that our results are coordinate-free, we show that the 'distortion factor' between the geodesic distances computed along two different manifolds $\mathcal{M}_1$ and $\mathcal{M}_2$ is uniformly bounded. The practical implication is that geodesic distances as measured in two different metrics can differ by no more than a constant factor, which precludes any situation where a system is stable in one metric (geodesic distances between trajectories shrink to zero) and not stable in another metric (geodesic distances do not shrink to zero).

**Theorem 3.** *Consider two Riemannian metrics* $\mathbf{M}_1(\boldsymbol{\theta}, t)$ *and* $\mathbf{M}_2(\boldsymbol{\theta}, t)$ *satisfying*

$$\alpha_1 \mathbf{I} \preceq \mathbf{M}_1(\boldsymbol{\theta}, t) \preceq \beta_1 \mathbf{I} \tag{10}$$

$$\alpha_2 \mathbf{I} \preceq \mathbf{M}_2(\boldsymbol{\theta}, t) \preceq \beta_2 \mathbf{I} \tag{11}$$

*with* $\alpha_i, \beta_i > 0$. *Then the corresponding geodesic distances evaluated between two points,* $\boldsymbol{\theta}$ *and* $\mathbf{y}$, *satisfy the bound*

$$\sqrt{\frac{\alpha_1}{\beta_2}} \leq \frac{d_{\mathcal{M}_1}(\boldsymbol{\theta}, \mathbf{y})}{d_{\mathcal{M}_2}(\boldsymbol{\theta}, \mathbf{y})} \leq \sqrt{\frac{\beta_1}{\alpha_2}}$$

Note that the lower bound on the metric follows from the requirement that to define a proper metric, the matrix $\mathbf{M}(\boldsymbol{\theta}, t)$ must be uniformly positive definite for all $\boldsymbol{\theta}$ and $t$. The upper bound can be ensured when e.g., the norm of the metric is Lipshitz with respect to the state, and the state remains within a finite set (as is the case with contracting optimizers, as we will see).

## 2 Main Results

### 2.1 Contracting Optimizers are Algorithmically Stable

In this section we prove our main result for continuous-time optimizers using the entire training batch. We start with this case because it is the simplest. Later on, we provide the same result for stochastic, discrete-time optimizers such as mini-batch stochastic gradient descent. We assume that our parameter update is *sum-separable* with respect to training set $S$

$$\dot{\boldsymbol{\theta}}_S = \mathbf{G}(\boldsymbol{\theta}_S, S) = \frac{1}{n} \sum_{i=1}^{n} \mathbf{g}(\boldsymbol{\theta}_S, z_i) \tag{12}$$

In this case the output of algorithm $\mathcal{A}(S)$ is the vector $\boldsymbol{\theta}_S$ obtained by simulating (12) for time $t$. We also assume that $||\mathbf{g}|| \leq \xi$, for some constant $\xi$. If we interpret $\mathbf{g}$ as the gradient of some loss $\ell$, then this corresponds to assuming that $\ell$ is $\xi$-Lipschitz. Finally, we assume that the optimizer is always initialized–perhaps randomly– within a ball of radius $C/2$ around some reference point. Now consider the same parameter update with respect to training set $S'$, which differs from $S$ in one example

$$\dot{\boldsymbol{\theta}}_{S'} = \mathbf{G}(\boldsymbol{\theta}_{S'}, S') \tag{13}$$

We can now state our first main result.

**Theorem 4.** *[Contraction Implies Algorithmic Stability] If the dynamics* (12) *are contracting in metric* $\mathbf{M} = \mathbf{T}(\boldsymbol{\theta}, t)^T \mathbf{T}(\boldsymbol{\theta}, t)$ *with rate* $\lambda$, *then* $\mathcal{A}$ *is uniformly* $\epsilon$-*stable, with*

$$\epsilon \leq \chi L e^{-\lambda t} C + \frac{2\chi L \xi}{\lambda n} \tag{14}$$

*where* $\chi$ *denotes a uniform upper-bound on the condition number of* $\mathbf{T}(\boldsymbol{\theta}, t)$. *Going forward we refer to* $\epsilon_{stab} \equiv \frac{2\chi L \xi}{\lambda n}$.

*Proof.* We shall write (13) as a perturbed version of (12) and then apply the robustness property of contracting systems to yield the result. Letting $k$ denote the index of the replaced element in $\mathcal{S}'$, we can write $\dot{\boldsymbol{\theta}}_{S'}$ as

$$\dot{\boldsymbol{\theta}}_{S'} = \frac{1}{n} \sum_{i=1}^{n} \mathbf{g}(\boldsymbol{\theta}_{S'}, z_i) - \frac{1}{n} \left[ \mathbf{g}(\boldsymbol{\theta}_{S'}, z_k) - \mathbf{g}(\boldsymbol{\theta}_{S'}, z_k') \right]$$

where we have just subtracted out the term involving $z_k$ from the sum, and added in the replacement term $z'_k$. This may be viewed as a perturbed version of (12), with disturbance

$$||\mathbf{d}(\boldsymbol{\theta}_{S'}, z_k, z'_k, n)|| = ||\frac{1}{n}(\mathbf{g}(\boldsymbol{\theta}_{S'}, z_k) - \mathbf{g}(\boldsymbol{\theta}_{S'}, z'_k))|| \leq \frac{2\xi}{n} = D \qquad (15)$$

Plugging $D$ into (9), multiplying through by $L$ because of (2), and taking the expectation $\mathbb{E}_{\mathcal{A}}$ to produce $R(0)$ yields the result. $\qquad \square$

**Remark 1** (Leave-One-Out Stability). *As pointed out in (Bousquet et al., 2020), for interpolation algorithms (such as, e.g., the highly overparameterized searches common in deep learning) it is more meaningful to analyze* leave-one-out *stability, rather than replace-one stability as we just did. The same dynamical robustness argument applies immediately to this case, with $D$ and therefore $\epsilon_{stab}$ reduced by a factor of two, as (15) is replaced by*

$$||\mathbf{d}(\boldsymbol{\theta}_{S'}, z_k, z'_k, n)|| = ||\frac{1}{n}\mathbf{g}(\boldsymbol{\theta}_{S'}, z_k)|| \leq \frac{\xi}{n}$$

**Remark 2** (Generalization with High Probability). *A well-known limitation of using algorithmic stability to derive generalization bounds is that the bounds only hold in expectation. However, one can use Chebyshev's inequality to derive generalization bounds that hold with high probability (Bousquet & Elisseeff, 2002; Elisseeff et al., 2005; Feldman & Vondrak, 2019; Bousquet et al., 2020). It is well known that these bounds are tight in the case when algorithmic stability scales with $1/n$, see e.g., Theorem 12 and Remark 13 in (Bousquet & Elisseeff, 2002). Theorem 4 shows that (after exponentially decaying transients) deterministic contracting optimizers generalize with rate $1/n$. In Section 2.2, Theorem 5 will show that this $1/n$ scaling also holds for the stochastic optimization case.*

**Remark 3** (Scaling Dynamics Does not Change Generalization Rate). *Note that if we 'speed up' the dynamics in (12) by some factor $\kappa > 0$*

$$\mathbf{G}(\boldsymbol{\theta}_S, S) \rightarrow \kappa\mathbf{G}(\boldsymbol{\theta}_S, S)$$

*one might intuitively expect the contraction rate to be scaled by $\kappa$ as well ($\lambda \rightarrow \kappa\lambda$), which would allow an arbitrary increase of the rate of generalization in (14) by simply increasing $\kappa$. Note however that this is prevented by the presence of $\xi$ in (14), which is also scaled by $\kappa$. The $\kappa$ terms in the numerator and denominator therefore cancel out, leaving $\epsilon_{stab}$ unchanged. The exponentially decaying term in (14), however, decays with new rate $\kappa\lambda$.*

**Remark 4** (Lipschitz Assumption). *As pointed out in Hardt et al. (2016), there are cases where $L$ as defined in (2) may not exist. For example, strongly convex functions have unbounded gradients on $\mathbb{R}^m$. In this case we will overload the symbol $L$ to be*

$$L = \sup_{\boldsymbol{\theta} \in \Omega} \sup_{z} ||\nabla\ell(\boldsymbol{\theta}, z)||_2$$

*where $\Omega$ denotes a compact set where the iterates of the optimizer are known to remain when initialized in a given compact region. For contracting optimizers and $\beta$-smooth loss functions ($\nabla^2\ell \preceq \beta\mathbf{I}$), $L$ is always finite. In particular, if we have some compact set of initial conditions and our optimizer is contracting (in a uniformly positive-definite metric), then the trajectories of the optimizer from any of those initial conditions will remain bounded. Indeed, any one trajectory will converge to a fixed point, and all the others must remain in a tube around its iterates (Lohmiller & Slotine, 1998). With this construction, we have a direct bound on the diameter of the set that the iterates of the optimizer must remain within, which we denote $\mathrm{diameter}(\Omega)$. In this case we have $L \leq \beta\,\mathrm{diameter}(\Omega)$.*

## 2.2 Stochastic, Contracting Optimizers are Algorithmically Stable

In this section we show that a variant of Theorem 4 holds for stochastic, discrete-time optimizers (for example mini-batch stochastic gradient descent). Consider the iterative optimizer

$$\boldsymbol{\theta}_{t+1}^S = \frac{1}{b}\sum_{i=1}^{b}\mathbf{g}(\boldsymbol{\theta}_t^S, z_i) \qquad (16)$$

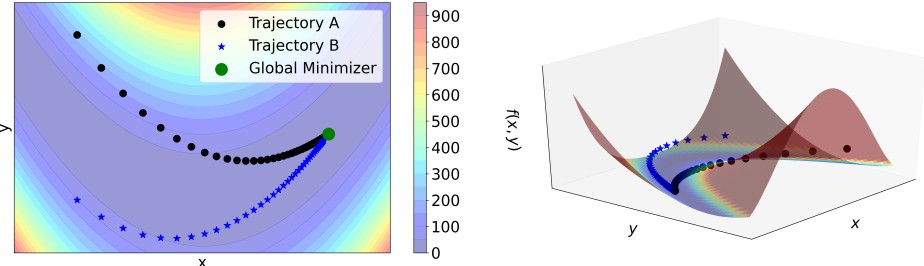

Figure 3: *Left subplot)* Two trajectories of a contracting optimizer, seeded from two different initial conditions, evolving over a non-convex loss surface (Rosenbrock function, $f(x,y) = 100(x^2 - y)^2 + (x - 1)^2$). Both exponentially converge to the global minimizer of the function. Trajectories superimposed over a contour plot of the loss surface. *Right subplot)* A different view of the same optimization process, more clearly displaying the non-convexity of the loss surface.

where $1 \leq b \leq n$ is the size of the mini-batch and $z_i$ are samples drawn randomly from set $S$. As before we assume that $\mathbf{g}$ is smooth and bounded as $||\mathbf{g}|| \leq \xi$. Since (16) defines a discrete-time, random dynamical system (Tabareau & Slotine, 2013) we have to define what we mean by 'contraction'. In particular we will rely on an assumption of 'contraction in expectation', by which we mean the following. Consider two instantiations of the same discrete-time, random dynamical system

$$\boldsymbol{\theta}_{t+1} = \mathbf{f}(\boldsymbol{\theta}_t, t, \Gamma)$$
$$\mathbf{y}_{t+1} = \mathbf{f}(\mathbf{y}_t, t, \Gamma)$$

with potentially different $\boldsymbol{\theta}_0$ and $\mathbf{y}_0$, and where $\Gamma$ denotes a *particular realization* of a stochastic process which is the same for both $\boldsymbol{\theta}$ and $\mathbf{y}$. In our case, this stochasticity stems from the random sampling of training set datapoints to form a mini-batch. We will say that this system is *contracting in expectation* if for a sequence of metrics $\mathbf{M}_0, \ldots, \mathbf{M}_t$ we have

$$\mathbb{E}_{\mathcal{A}}[d_{\mathcal{M}_{t+1}}(\mathbf{f}(\boldsymbol{\theta}_t, t, \Gamma), \mathbf{f}(\mathbf{y}_t, t, \Gamma))] \leq \mu \mathbb{E}_{\mathcal{A}}[d_{\mathcal{M}_t}(\boldsymbol{\theta}_t, \mathbf{y}_t)]$$

where $0 < \mu < 1$ and each metric is bounded $M_{min}\mathbf{I} \preceq \mathbf{M}_i \preceq M_{max}\mathbf{I}$. We can now state the following theorem

**Theorem 5.** *[Contraction Implies Algorithmic Stability (Stochastic, Discrete)] Assume* (16) *is contracting in expectation, as defined above. In this case $\mathcal{A}$ is uniformly $\epsilon$-stable with bound*

$$\epsilon \leq L\chi C\mu^t + \frac{2\chi L\xi}{(1-\mu)n} \tag{17}$$

*Proof.* See appendix section A.1. □

**Remark 5.** *Note that* (17) *does not depend on the batch size, $b$. Intuitively, this is because the resampled datapoint $z'_k$ will appear in a given batch with probability $b/n$, but the magnitude of the disturbance this causes on the optimizer dynamics scales with $1/b$. These two terms interact multiplicatively, so that the $b$ terms cancel, leaving only the $1/n$ scaling.*

## 3   Examples of Metrics

### 3.1   Preconditioned Gradient Descent On Strongly Convex Loss Functions

In this example we show that our theory reproduces known stability bounds for gradient descent on strongly convex losses. To illustrate the role of the contraction metric, we consider *preconditioned* gradient descent. Consider this descent over an empirical loss function which is $\gamma$-strongly convex with respect to a parameter vector $\boldsymbol{\theta}$

$$\dot{\boldsymbol{\theta}} = -\mathbf{P}^{-1}\nabla\mathcal{L}$$

where $\mathbf{P}$ is a positive-definite and symmetric matrix. Denote the largest and smallest eigenvalues of $\mathbf{P}$ as $p_{max}$ and $p_{min}$, respectively. The Jacobian of this system is

$$\mathbf{J} = \frac{\partial \dot{\boldsymbol{\theta}}}{\partial \boldsymbol{\theta}} = -\mathbf{P}^{-1}\nabla^2 \mathcal{L}$$

Picking the metric $\mathbf{M} = \mathbf{P}$, we see that

$$\mathbf{PJ} + \mathbf{J}^T\mathbf{P} = -2\nabla^2\mathcal{L} \leq -2\gamma\mathbf{I} \leq -\frac{2\gamma}{p_{max}}\mathbf{P}$$

and thus the system is contracting in metric $\mathbf{P}$ with rate $\lambda = \gamma/p_{max}$. Our algorithmic stability bound is therefore

$$\epsilon_{stab} = \sqrt{\frac{p_{max}^3}{p_{min}}}\frac{2L^2}{\gamma n}$$

Where $L$ is given by (4). Note that in the case of regular gradient descent, without preconditioning (i.e., $\mathbf{P} = \mathbf{I}$) the above analysis shows that $\lambda = \gamma$ and $\chi = 1$. Plugging these numbers into equation (14) yields the following

$$\epsilon_{stab} = \frac{2L^2}{\gamma n}$$

which is precisely the result of Theorem 3.9 in (Hardt et al., 2016).

**Remark 6** (Natural Gradient on Geodesically Strongly Convex Losses). *Natural gradients are a popular way to incorporate geometric information about the loss surface into gradient-based optimization techniques (Amari, 1998; Zhang et al., 2019). An equivalence between g-Strong Convexity and global contraction of natural gradient flows was given in (Theorem 1, (Wensing & Slotine, 2020)). That is, the optimizer dynamics*

$$\dot{\boldsymbol{\theta}} = -\mathbf{M}(\boldsymbol{\theta})^{-1}\nabla\mathcal{L}(\boldsymbol{\theta})$$

*are globally contracting in the metric $\mathbf{M}$ if and only if $\mathcal{L}(\boldsymbol{\theta})$ is geodesically strongly convex over $\mathcal{M}$. In this case Theorem 4 of the present work applies immediately, in precisely the same fashion as the preceding subsection.*

## 3.2 The Choice of Metric is Critical

We now consider a simple two-dimensional, analytical example where failing to include a metric leads to inconclusive stability and generalization bounds. Consider minimizing the classical nonconvex Rosenbrock function (Figure 3)

$$\ell_i = a_i(\theta_1^2 - \theta_2)^2 + (\theta_1 - 1)^2$$

where $\boldsymbol{\theta} = \begin{bmatrix} \theta_1 & \theta_2 \end{bmatrix}^T$ and $a_i$ is a bounded random variable whose expected value is 100. The mean loss over training samples is

$$\mathcal{L} = \frac{1}{n}\sum_{i=1}^{n}\ell_i = \langle a \rangle(\theta_1^2 - \theta_2)^2 + (\theta_1 - 1)^2$$

where $\langle a \rangle$ denotes the empirical average

$$\langle a \rangle = \frac{1}{n}\sum_{i=1}^{n}a_i$$

We consider the large $n$ limit, where $\langle a \rangle \approx 100$. It was shown in Wensing & Slotine (2020) that when $a = 100$, the natural gradient descent dynamics

$$\frac{\mathrm{d}}{\mathrm{d}t}\boldsymbol{\theta} = -\mathbf{M}^{-1}(\boldsymbol{\theta})\nabla_{\boldsymbol{\theta}}\mathcal{L}$$

is contracting with rate $\alpha > 0$ in metric

$$\mathbf{M}(\boldsymbol{\theta}) = \begin{bmatrix} 400\,\theta_1^2 + 1 & -200\,\theta_1 \\ -200\,\theta_1 & 100 \end{bmatrix}$$

By Theorem 4, this implies a generalization rate of order $1/n\alpha$. We can now ask what happens if we use the identity metric in the stability analysis, instead of $\mathbf{M}(\boldsymbol{\theta})$. It can be shown that the Jacobian of the natural gradient descent dynamics above are

$$\mathbf{J} = \frac{\partial \dot{\boldsymbol{\theta}}}{\partial \boldsymbol{\theta}} = 2 \begin{bmatrix} 1 & 0 \\ \theta_1 - 1 & -1 \end{bmatrix}$$

without the metric, the contraction condition (4) requires that the eigenvalues of the symmetric part of $\mathbf{J}$ are uniformly negative definite. However, it is readily shown these eigenvalues are

$$\lambda_1 = -\sqrt{\theta_1^2 - 2\theta_1 + 5} = -\lambda_2$$

Because $\lambda_2$ is always positive, the natural gradient descent dynamics are not contracting in the identity metric, and thus one would not be able to derive generalization bounds in this case.

### 3.3 Identity Metric is Optimal for Ridge Regression

In section 3.2, we demonstrated that the choice of metric can be critical for establishing stability and generalization bounds. For certain nonlinear optimization problems, this metric will naturally be state-dependent (such as in natural gradient descent). However, for other optimization problems, in particular *linear* ones, a constant metric is to be expected. In this setting, one can ask if there exists an *optimal metric* in terms of providing the best generalization bounds. In the case of ridge regression, we prove that the answer is yes, and that this metric is in fact the identity metric. This is sharply contrasted with nonlinear optimization problems, where the identity metric can fail to produce stability and generalization bounds (see section 3.2).

For constant metrics, our stability bound $\epsilon_{stab} \sim \frac{\chi}{\lambda}$ depends on the condition number of the contraction metric (specifically its square root) to the contraction rate *measured in that metric*. Different metrics yield different $\epsilon_{stab}$, so it is natural to ask whether an 'optimal' metric $\mathbf{M}_{optimal}$ exists, such that

$$\epsilon_{stab}(\mathbf{M}_{optimal}) \leq \epsilon_{stab}(\mathbf{M})$$

While finding such a metric is in general not easy to do, we show that it is possible in the case of gradient descent for kernel ridge regression (Shawe-Taylor et al., 2004). Kernel methods (which are inherently linear) can be used to derive insights into nonlinear systems such deep neural networks (Jacot et al., 2018; Lee et al., 2019; Fort et al., 2020; Canatar et al., 2021). Without loss of generality, we assume an element-wise feature map such that for a matrix $\mathbf{X} \in \mathbb{R}^{q \times z}$, the matrix $\phi(\mathbf{X}) \in \mathbb{R}^{q \times z}$ satisfies $\phi(\mathbf{X})_{ij} = \phi(\mathbf{X}_{ij})$. The squared-loss for kernel ridge-regression is

$$\mathcal{L} = \frac{1}{2n} \sum_{i=1}^{n} (\phi(\mathbf{x}_i)\mathbf{w} - y_i)^2 + \frac{\alpha}{2} ||\mathbf{w}||^2$$

where the $\phi(\mathbf{x}_i)$ are feature row vectors, $\mathbf{w} \in \mathbb{R}^m$ are the linear model parameters to be learned, and the $y_i \in \mathbb{R}$ are target labels. The parameter $\alpha > 0$ is the regularization parameter. Under gradient descent $\dot{\mathbf{w}} = -\nabla \mathcal{L}$ the Jacobian of the optimizer dynamics is

$$\frac{\partial \dot{\mathbf{w}}}{\partial \mathbf{w}} = \mathbf{J} = -(\mathbf{G} + \alpha \mathbf{I})$$

where $\mathbf{G} \equiv \frac{1}{n}\phi(\mathbf{X})^T \phi(\mathbf{X})$ and $\mathbf{X} \in \mathbb{R}^{n \times d}$ is a constant matrix with $\mathbf{x}_i$ as the $i^{th}$ row. The Jacobian $\mathbf{J}$ is symmetric, constant, and negative-definite. Thus, the optimizer is contracting in the identity metric with rate $\lambda_I = \lambda_{min}(\mathbf{G}) + \alpha$, where $\lambda_{min}(\cdot)$ denotes the smallest eigenvalue. We now prove the following

**Theorem 6.** *For kernel ridge regression, the algorithmic stability bound $\epsilon_{stab}$ is minimized for $\mathbf{M} = \mathbf{I}$.*

*Proof.* Recall that for an arbitrary, constant metric we are looking for a positive-definite symmetric $\mathbf{Q}$ such that

$$\mathbf{MJ} + \mathbf{JM} = -\mathbf{Q} \leq -2\lambda\mathbf{M}$$

Ignoring $\chi$ for a moment, we can ask: out of the set of all possible metrics, is there a metric that yields the *largest* contraction rate $\lambda$? An interesting result from linear dynamical systems theory is that the answer is in fact yes. While there can be many metrics for linear systems that give the largest possible $\lambda$, one can always be found from setting $\mathbf{Q} = \mathbf{I}$ and solving for $\mathbf{M}$ (see, e.g., section 3.5.5 in (Slotine & Li, 1991)). Since $\mathbf{J}$ is symmetric, in our case this metric corresponds to the diagonalizing metric

$$\mathbf{M}_{largest} = \frac{1}{2}\mathbf{J}^{-1} = \frac{1}{2}(\mathbf{G} + \alpha\mathbf{I})^{-1}$$

The contraction rate $\lambda_{largest}$ corresponding to this metric is

$$\lambda_{largest} = \frac{1}{2}\frac{1}{\lambda_{max}(\mathbf{M}_{largest})} = \lambda_{min}(\mathbf{G}) + \alpha$$

which is precisely the same contraction rate as measured in the identity metric. Thus $\lambda_I = \lambda_{largest}$. Now we simply use the fact that $\chi_I = 1 \leq \chi_M$ for any metric. Since $\mathbf{M} = \mathbf{I}$ corresponds to the largest possible $\lambda$ and the smallest possible $\chi = 1$, the ratio of $\chi$ to $\lambda$ is minimal over all possible $\mathbf{M}$ when $\mathbf{M} = \mathbf{I}$. Thus

$$\epsilon_{stab}(\mathbf{I}) \leq \epsilon_{stab}(\mathbf{M})$$

$\square$

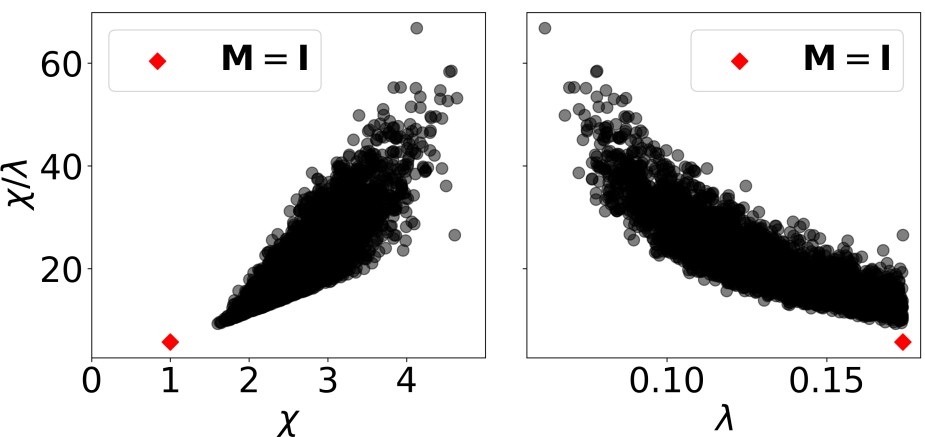

Figure 4: For a fixed $\mathbf{G} \in \mathbb{R}^{3\times3}$, we randomly generate $\mathbf{Q}$ and solve for $\mathbf{M}$. We then calculate the ratio $\chi/\lambda_M$ (the only metric-dependent terms in our algorithmic stability bound). We repeat this procedure 4000 times. *Left subplot)* The ratio $\chi/\lambda_M$ plotted against $\chi$. *Right subplot)* The ratio $\chi/\lambda_M$ bound plotted against $\lambda_M$. Details are in the main text. These plots illustrate our theoretical result that the identity metric gives the optimal (i.e smallest) algorithmic stability bound. They also illustrate the reason: the identity metric simultaneously obtains the smallest condition number and the largest contraction rate, thus minimizing the ratio of the former to the latter.

This result is illustrated in Figure 4. To create this plot we generated a random $\mathbf{G} \in \mathbb{R}^{3\times3}$. Then we generated random $\mathbf{Q}$ and solved the Lyapunov equation for $\mathbf{M}$ using an implementation of the Bartels-Stewart algorithm in SciPy (Bartels & Stewart, 1972; Virtanen et al., 2020). In addition to these random $\mathbf{Q}$, we also set $\mathbf{Q} = \mathbf{I}$ to obtain the $\mathbf{M}$ corresponding to the largest $\lambda$. For each of these $\mathbf{Q}$ and $\mathbf{M}$ pairs, $\lambda_M$ is given by $\lambda_M = \frac{1}{2}\frac{\lambda_{min}(\mathbf{Q})}{\lambda_{max}(\mathbf{M})}$ (Slotine & Li, 1991). We computed $\mathbf{T}$ via a Cholesky decomposition (also using SciPy) and then performed a singular value decomposition to obtain $\chi$. One interpretation of this result is: there is no 'better' coordinate system. That is, there is no coordinate transformation we could perform on the state vector $\mathbf{w}$ which would give us tighter algorithmic stability bounds. This is because a constant metric $\mathbf{M} = \mathbf{T}^T\mathbf{T}$ corresponds to the coordinate change $\mathbf{w} \rightarrow \mathbf{Tw}$.

### 3.4 Immersions and Embeddings

Early concepts in machine learning, such as support vector machines (Vapnik, 1999) and kernel methods (Shawe-Taylor et al., 2004), build on the idea that immersing data points into a *higher* dimensional space can often simplify learning. Using the same intuition, we explore the potential to simplify the contraction analysis of an optimizer through the use of a higher-dimensional immersion. Contraction properties of immersions in higher-dimensional spaces can be easily translated in terms of metrics in the original low dimensional space, as pointed out in (Yi & Manchester, 2021) in the special case of linear Koopman embeddings. Conversely, modern machine learning makes extensive use of compressed representations and *lower* dimensional embeddings of the original space (Goodfellow et al., 2016), e.g. through the use of autoencoders. We show how contraction properties in the compressed (or latent) space can be related to contraction properties of the original space. In both cases, the generalization bounds derived in this paper can be readily applied.

**Immmersions** Consider immersing a lower dimensional state $\boldsymbol{\theta} \in \mathbb{R}^m$ into a higher dimensional one $\mathbf{z} \in \mathbb{R}^d$, with $d \geq m$

$$\mathbf{z} = \psi(\boldsymbol{\theta}) \tag{18}$$

where $\psi(\boldsymbol{\theta})$ is continuously differentiable, $\frac{\partial \psi}{\partial \boldsymbol{\theta}}$ has full column rank, and the vector fields $\dot{\mathbf{z}}$ and $\dot{\boldsymbol{\theta}}$ are compatible (i.e., for each $\boldsymbol{\theta}$, $\dot{\mathbf{z}}$ at $\psi(\boldsymbol{\theta})$ agrees with $\frac{\partial \psi}{\partial \boldsymbol{\theta}}\dot{\boldsymbol{\theta}}$ at $\boldsymbol{\theta}$).

Assume that the higher dimensional system is contracting in some metric $\mathbf{M}(\mathbf{z}, t)$

$$\frac{d}{dt} \; \delta\mathbf{z}^T \mathbf{M}(\mathbf{z},t)\delta\mathbf{z} \;\; \leq \;\; -2\lambda \; \delta\mathbf{z}^T \mathbf{M}(\mathbf{z},t)\delta\mathbf{z} \tag{19}$$

with $\delta\mathbf{z}$ a differential displacement (Lohmiller & Slotine, 1998) and $\lambda$ a strictly positive constant. Since

$$\delta\mathbf{z} = \frac{\partial\psi}{\partial\boldsymbol{\theta}} \; \delta\boldsymbol{\theta}$$

equation (19) can be equivalently written

$$\frac{d}{dt} \; \delta\boldsymbol{\theta}^T \mathbf{M}_\theta(\boldsymbol{\theta},t)\delta\boldsymbol{\theta} \;\; \leq \;\; -2\lambda \; \delta\boldsymbol{\theta}^T \mathbf{M}_\theta(\boldsymbol{\theta},t)\delta\boldsymbol{\theta}$$

with

$$\mathbf{M}_\theta(\boldsymbol{\theta},t) \;\; = \;\; \frac{\partial\psi}{\partial\boldsymbol{\theta}}^T \mathbf{M}(\psi(\boldsymbol{\theta}),t) \; \frac{\partial\psi}{\partial\boldsymbol{\theta}}$$

This shows that the original $\boldsymbol{\theta}$ system is also contracting, in metric $\mathbf{M}_\theta$ , with the same contraction rate $\lambda$.

More generally, one can consider a *differential immersion* of the form

$$\delta\mathbf{z} = \mathbf{T}(\boldsymbol{\theta},t)\delta\boldsymbol{\theta}$$

with $\mathbf{T}(\boldsymbol{\theta},t)$ of full column rank, and assuming as before that $\frac{d}{dt}\delta\mathbf{z}$ and $\frac{d}{dt}\delta\boldsymbol{\theta}$ are compatible. Note that such an immersion does not require the above equation to be integrable, i.e., an explicit $\mathbf{z}$ to exist as in (18). It corresponds to immersing the vector field for $\boldsymbol{\theta}$ into one defined in some higher dimensional Riemannian manifold rather than a Euclidean space. By the same argument as above, contraction of the higher-dimensional system in the metric $\mathbf{M}(\psi(\boldsymbol{\theta}),t)$ implies contraction of the lower dimensional system at the same rate in the metric

$$\mathbf{M}_\theta(\boldsymbol{\theta},t) \;\; = \;\; \mathbf{T}^T(\boldsymbol{\theta},t) \; \mathbf{M}(\psi(\boldsymbol{\theta}),t) \; \mathbf{T}(\boldsymbol{\theta},t)$$

This result may simplify the contraction analysis of the lower dimensional one. In particular, if the higher dimensional system is contracting in a *constant* metric $\mathbf{M}$, so that $\dot{\mathbf{M}} = \mathbf{0}$ in (4), then we can immediately conclude that the lower dimensional system is contracting in the time-varying, state-dependent metric $\mathbf{M}_\theta(\boldsymbol{\theta},t) \;\; = \;\; \mathbf{T}^T(\boldsymbol{\theta},t) \; \mathbf{M} \; \mathbf{T}(\boldsymbol{\theta},t)$ , without having to deal explicitly with the time-derivative of that metric.

In the generalization bound of Theorem (4), both the contraction rate $\lambda$ and the $1/n$ scaling remain unchanged. In addition, decomposing $\mathbf{M}(\psi(\boldsymbol{\theta}),t)$ as $\mathbf{M}(\psi(\boldsymbol{\theta}),t) = \mathbf{B}^T(\psi(\boldsymbol{\theta}),t)\mathbf{B}(\psi(\boldsymbol{\theta}),t)$, $\chi$ is now replaced by $\chi = \chi_T\chi_B$ where $\chi_T$ and $\chi_B$ denote upper bounds on the condition numbers of $\mathbf{T}(\boldsymbol{\theta},t)$ and $\mathbf{B}(\psi(\boldsymbol{\theta}),t)$.

**Embeddings** The same logic can in fact be applied to embeddings in lower-dimensional spaces. Assume for instance that there exists a high-quality *autoencoder* for the original state $\boldsymbol{\theta} \in \mathbb{R}^m$. In other words, there exists an encoder $E$ and a decoder $D$ such that $\mathbf{z} = E(\boldsymbol{\theta}) \in \mathbb{R}^d$ and (ideally) $D(\mathbf{z}) = \boldsymbol{\theta}$, with $d \le m$. Assume the compressed (latent) $\mathbf{z}$ dynamics are contracting with rate $\lambda$ in a metric of a specific form,

$$\frac{d}{dt} \ \delta\mathbf{z}^T\mathbf{M}_z(\mathbf{z},t)\delta\mathbf{z} \ \le \ -2\lambda \ \delta\mathbf{z}^T\mathbf{M}_z(\mathbf{z},t)\delta\mathbf{z} \qquad \text{with} \qquad \mathbf{M}_z(\mathbf{z},t) = \frac{\partial D}{\partial \mathbf{z}}^T \mathbf{M}(\mathbf{z},t) \ \frac{\partial D}{\partial \mathbf{z}}$$

then the original $\boldsymbol{\theta}$ dynamics are contracting in metric $\mathbf{M}(E(\boldsymbol{\theta}),t)$ with the same rate $\lambda$. This follows directly from substituting above the identity $\delta\boldsymbol{\theta} = \frac{\partial D}{\partial \mathbf{z}} \ \delta\mathbf{z}$ and using $\mathbf{z} = E(\boldsymbol{\theta})$. One may choose to compute generalization bounds in the latent space or the original space.

# 4 Weaker Notions of Stability

Contraction imposes a strong condition on trajectories: they must converge toward one another exponentially. Such convergence can be expected for parameter trajectories around isolated local or global minima, as discussed above. However, in modern machine learning, one often observes parameter trajectories which converge towards a common basin of low/zero loss, where minima may lie among a low-dimensional manifold (Garipov et al., 2018; Draxler et al., 2018; Fort et al., 2020; Liu et al., 2021). To accommodate these cases, we now discuss several weaker notions of contraction–specifically local contraction, semi-contraction, partial contraction, and output contraction–which also yield 'well-behaved' algorithmic stability bounds.

## 4.1 Loss Surfaces with Many Local Minima

A contraction region (i.e., a forward invariant region of state space that satisfies Definition 2) for an autonomous system contains at most one equilibrium point (Lohmiller & Slotine, 1998). From this it follows that gradient descent over a loss surface with many local equilibria cannot be globally contracting. Fortunately, if Definition 2 holds within a subset of state-space, and additionally the system can be shown to remain in that subset for all time (i.e., the subset is forward invariant), then that system is locally contracting. This motivates the following general result, as well as an optimization-specific remark.

**Theorem 7.** *Consider the system* (3) *initialized inside an inner Euclidean ball of radius $b$, which is fully contained within an outer contraction region (which we also assume without loss of generality to be a Euclidean ball) of radius $B > b$. Assume that* (3) *stays within the inner ball for all time. Now consider the perturbed dynamics* (5). *If $B \ge b(\chi + 1) + \frac{\chi D}{\lambda}$ then* (5) *stays within the outer contraction region for all time, and the robustness result* (7) *holds.*

*Proof.* By (7), the perturbed trajectory will be at most distance $\chi b + \frac{\chi D}{\lambda}$ from the unperturbed trajectory. Since the unperturbed trajectory is always contained within a ball of radius $b$, this implies the perturbed trajectory is also contained in a ball of radius $b + \chi b + \frac{\chi D}{\lambda}$, assuming it stays in a contraction region. To ensure that it does in fact stay within a contraction region, we must have that $B \ge b(\chi + 1) + \frac{\chi D}{\lambda}$. $\qquad\square$

**Remark 7.** *In the case of continuous-time optimizer* (12), *we have that $\boldsymbol{\theta}_S$ converges exponentially to the equilibrium point $\boldsymbol{\theta}_S^*$ enclosed by the contraction region*

$$||\boldsymbol{\theta}_S^* - \boldsymbol{\theta}_S|| \le \chi e^{-\lambda t} b$$

*Since $\boldsymbol{\theta}_S^*$ is a particular trajectory of the optimizer dynamics, by robustness we also have*

$$||\boldsymbol{\theta}_S^* - \boldsymbol{\theta}_{S'}|| \le \chi e^{-\lambda t} b + \frac{2\chi\xi}{\lambda n}$$

*by the triangle inequality*

$$||\boldsymbol{\theta}_S - \boldsymbol{\theta}_{S'}|| \le 2\chi e^{-\lambda t} b + \frac{2\chi\xi}{\lambda n}$$

*this puts the following lower bound on the size of the contraction region $B$*

$$B > 2b\chi + \frac{2\chi\xi}{\lambda n}$$

### 4.2 Semi-Contracting Optimizers

If the optimizer is not strictly contracting ($\lambda > 0$), but instead is *semi-contracting* (i.e., $\lambda \geq 0$) then our algorithmic stability bound $\epsilon_{stab}$ is not independent of the training time. This is because the geodesic distance $d_{\mathcal{M}}(\boldsymbol{\theta}, \boldsymbol{\theta}_p)$ between unperturbed and perturbed trajectories evolves according to

$$\frac{\mathrm{d}}{\mathrm{d}t} d_{\mathcal{M}}(\boldsymbol{\theta}, \boldsymbol{\theta}_p) + \lambda d_{\mathcal{M}}(\boldsymbol{\theta}, \boldsymbol{\theta}_p) \leq ||\mathbf{T}(\boldsymbol{\theta}, t)\mathbf{d}(\boldsymbol{\theta}_p, t)||$$

If the only information we have about $\lambda$ is that it is non-negative, then we can only bound the distance between trajectories as

$$d_{\mathcal{M}}(\boldsymbol{\theta}, \boldsymbol{\theta}_p) \leq \sup(||\mathbf{T}(\boldsymbol{\theta}, t)\mathbf{d}(\boldsymbol{\theta}_p, t)||)T + R(0)$$

Considering the disturbance bound $||\mathbf{d}(\boldsymbol{\theta}_p, t)|| \leq \frac{2\xi}{n}$ leads to the algorithmic stability bound

$$\epsilon \leq L \left[ \frac{2\chi\xi}{n} T + R(0) \right] \tag{20}$$

This bound holds generally for semi-contracting systems. However, without additional information about the optimizer dynamics, this bound gets worse as $T \to \infty$. If we know additional information about the dynamics–for example, that they are modulated by a decaying learning rate–much tighter bounds can be obtained. We show this with the following example.

### 4.2.1 Example: Gradient Flows on Convex Losses

Here we show how the above analysis reproduces a well-known result from Hardt et al. (2016) regarding the algorithmic stability of SGD on convex (but not strongly convex) losses.

Assume that the Hessian of the loss function is positive semi-definite

$$\nabla^2 \mathcal{L} \geq 0$$

and consider the gradient flow with learning rate scheduler (Goodfellow et al., 2016)

$$\dot{\boldsymbol{\theta}} = -\alpha(t)\nabla\mathcal{L}$$

where $\alpha(t) \geq 0$. This optimizer is semi-contracting in the identity metric, since

$$\frac{\partial\dot{\boldsymbol{\theta}}}{\partial\boldsymbol{\theta}} = -\alpha(t)\nabla^2\mathcal{L} \leq 0$$

In this case the disturbance term in Theorem 4 is the same as before, only with an additional $\alpha(t)$ factored in. To facilitate comparison with Hardt et al. (2016), we assume as they do that the optimizer is always initialized at the origin (i.e., $R(0) = 0$). The Euclidean distance between the optimizer trajectories on training sets $S$ and $S'$ evolves according to

$$\dot{R} \leq \alpha(t)D$$

where $D = 2L/n$. Integrating this inequality and setting $R(0) = 0$ yields the algorithmic stability bound

$$\epsilon \leq \frac{2L^2}{n} \int_{t=0}^{T} \alpha(t)dt$$

which is the result of Hardt et al. (2016), Theorem 3.8. We remind the reader that the extra factor of $L$ is picked up from (2). This result helps explain why a decaying learning rate is a useful strategy in deep learning–if the learning rate decays quickly enough (e.g., exponentially), the above integral converges, so that $n$ and $T$ do not compete with each other, as they do in (20).

**Remark 8.** *The equivalence between semi-contraction of natural gradient flows in the natural metric and geodesic convexity was recently proven in Wensing & Slotine (2020). Thus the above algorithmic stability bound extends immediately to this case.*

### 4.3 Output Contractions, Neural Tangent Kernels, and Polyak-Lojasiewicz

The same robustness arguments developed above can be applied to *outputs* of nonlinear models, through the concept of a Neural Tangent Kernel (NTK) (Jacot et al., 2018). While so far we have been using contraction of the model parameters $\boldsymbol{\theta}$ to derive generalization bounds, for this section will use the contraction of the model outputs to derive generalization bounds. NTK training may be viewed as contraction on a Hilbert space, a special case of Riemannian manifold, allowing a straightforward application of the robustness arguments developed above. As we will show, contraction of the model outputs is a consequence of using gradient descent/flow together with mean-squared loss, and does not require any additional assumptions, such as convexity of the loss with respect to the model parameters.

Consider the following nonlinear model $\mathcal{F}_{\boldsymbol{\theta}}$, parameterized by a set of weights $\boldsymbol{\theta} \in \mathbb{R}^m$. For each input vector $\mathbf{x}_i$ the model produces a $p$-dimensional output. We denote this output $\mathbf{y}_i$. We will also find it useful to define a vector $\overline{\mathbf{y}}$ obtained by stacking these vectors

$$\mathbf{y}_i = \mathcal{F}_{\boldsymbol{\theta}}(\mathbf{x}_i) = \mathcal{F}(\mathbf{x}_i) \in \mathbb{R}^p \quad \text{and} \quad \overline{\mathbf{y}} = \begin{bmatrix} \mathbf{y}_1 \\ \vdots \\ \mathbf{y}_n \end{bmatrix} \in \mathbb{R}^{np} \tag{21}$$

We will assume that $\mathcal{F}_{\boldsymbol{\theta}}(\mathbf{x}_i)$ is Lipschitz with respect to the weights $\boldsymbol{\theta}$, namely that there exists some $\kappa \geq 0$ such that

$$||\nabla_{\boldsymbol{\theta}} \mathcal{F}(\mathbf{x}_i)||_2 \leq \kappa$$

We will also assume that the model is sufficiently overparameterized (Liu et al., 2021; Arora et al., 2019; Nguyen et al., 2021) so that the NTK, $\mathbf{H} \in \mathbb{R}^{np \times np}$, is positive definite

$$\mathbf{H} = \frac{1}{n} \nabla_{\boldsymbol{\theta}}^T \overline{\mathbf{y}} \, \nabla_{\boldsymbol{\theta}} \overline{\mathbf{y}} \succeq \lambda_0 \quad \text{with} \quad \lambda_0 > 0$$

With these assumptions in hand, the main result of this section is as follows.

**Theorem 8.** *Using gradient flow together with mean-squared loss, the overparameterized model defined in (21) is algorithmically stable with rate*

$$\epsilon_{stab} \sim \mathcal{O}\left( \frac{\kappa}{\lambda_0 \sqrt{n}} \right)$$

*where $\lambda_0 > 0$ is a uniform lower bound on the smallest eigenvalue of the Neural Tangent Kernel.*

*Proof.* As in the previous sections, we will consider training this model on two datasets $\mathcal{S}$ and $\mathcal{S}'$ which differ at a single datapoint at index $k$. We will focus on the case where the parameters $\boldsymbol{\theta}$ of the model defined in (21) are trained using gradient flow. To distinguish between the two different models, we will use the notation $\mathbf{y}_i$ and $\boldsymbol{\theta}_{\mathcal{S}}$ when referring to the model learned by training on dataset $\mathcal{S}$, and the notation $\mathbf{y}'_i$ and $\boldsymbol{\theta}_{\mathcal{S}'}$ when referring to the model learned by training on dataset $\mathcal{S}'$. Similarly, to refer to the replaced datapoint we will use the notation $\mathbf{x}'_k$ to denote the input and $\mathbf{y}'_k$ to denote the desired output. To begin, we note that (1) is agnostic with respect to what we define as the outputs of the optimization algorithm $\mathcal{A}$. In this section, we consider the outputs of the trained model defined in (21) as the outputs of the optimization procedure. That is

$$\mathbf{y}_i(t \to \infty) = \mathcal{A}(\mathcal{S})$$

as in (2), we will assume that the loss function is Lipschitz with respect to the model outputs

$$\forall i \quad \mathbb{E}_{\mathcal{A}}[|\ell(\mathbf{y}_i, z) - \ell(\mathbf{y}'_i, z)|] \leq L_y \, \mathbb{E}_{\mathcal{A}}||\mathbf{y}_i - \mathbf{y}'_i|| \tag{22}$$

The goal of this section is to show that as the number of training samples $n \to \infty$, the distance $||\mathbf{y}_i - \mathbf{y}'_i||$ shrinks to zero, which implies a vanishing generalization gap via (1). With this notation in hand, consider the time evolution of the output $\mathbf{y}_i$

$$\frac{\mathrm{d}}{\mathrm{d}t}\mathbf{y}_i = \nabla_{\boldsymbol{\theta}_{\mathcal{S}}} \mathcal{F}(\mathbf{x}_i)^T \frac{\mathrm{d}}{\mathrm{d}t}\boldsymbol{\theta}_{\mathcal{S}} = -\frac{1}{n} \sum_{j=1}^{n} \nabla_{\boldsymbol{\theta}_{\mathcal{S}}} \mathcal{F}(\mathbf{x}_i)^T \nabla_{\boldsymbol{\theta}_{\mathcal{S}}} \ell(\mathbf{y}_j, \mathbf{y}_j) \tag{23}$$

Using the same idea as in the proof of Theorem 4, we view the dynamics of $\mathbf{y}_i'$ as a perturbed version of the dynamics of $\mathbf{y}_i$.

$$\frac{\mathrm{d}}{\mathrm{d}t}\mathbf{y}_i' = \nabla_{\boldsymbol{\theta}_{\mathcal{S}'}}\mathcal{F}(\mathbf{x}_i)^T \frac{\mathrm{d}}{\mathrm{d}t}\boldsymbol{\theta}_{\mathcal{S}'} = -\frac{1}{n}\sum_{j=1}^{n} \nabla_{\boldsymbol{\theta}_{\mathcal{S}'}}\mathcal{F}(\mathbf{x}_i)^T \nabla_{\boldsymbol{\theta}_{\mathcal{S}'}}\ell(\mathbf{y}_j', \mathbf{y}_j) + \mathbf{d}_i(t) \tag{24}$$

where the disturbance term $\mathbf{d}_i(t)$ is given by

$$\mathbf{d}_i(t) = \frac{1}{n}\left[\nabla_{\boldsymbol{\theta}_{\mathcal{S}'}}\mathcal{F}(\mathbf{x}_k')^T \nabla_{\boldsymbol{\theta}_{\mathcal{S}'}}\ell(\mathbf{y}_i', \mathbf{y}_k') - \nabla_{\boldsymbol{\theta}_{\mathcal{S}'}}\mathcal{F}(\mathbf{x}_k)^T \nabla_{\boldsymbol{\theta}_{\mathcal{S}'}}\ell(\mathbf{y}_i', \mathbf{y}_k)\right]$$

with $k$ being the index of the replaced element in $\mathcal{S}'$. As in previous sections, the effect of this disturbance term is to "subtract out" the gradient corresponding to index $k$ in dataset $\mathcal{S}$, and to "add in" the gradient corresponding to index $k$ in dataset $\mathcal{S}'$. The norm of the disturbance term is upper-bounded simply as

$$||\mathbf{d}_i(t)|| \leq \frac{2\kappa L}{n} \tag{25}$$

Note that this disturbance term applies to each model output *separately*, whereas the NTK perspective emerges when considering all model outputs jointly and using the mean squared loss function. To obtain the final generalization bound using (25), we also have to consider all the outputs jointly and use the mean squared loss function. The time evolution of the stacked vector $\overline{\mathbf{y}}$ (21) is obtained by taking the time derivative and substituting in (23)

$$\frac{\mathrm{d}}{\mathrm{d}t}\overline{\mathbf{y}} = \begin{bmatrix} \frac{\mathrm{d}}{\mathrm{d}t}\mathbf{y}_1 \\ \vdots \\ \frac{\mathrm{d}}{\mathrm{d}t}\mathbf{y}_n \end{bmatrix} = -\mathbf{H}(t)\,(\overline{\mathbf{y}} - \overline{\mathbf{y}}_d) \quad \text{with} \quad \mathbf{H}(t) = \frac{1}{n}\nabla_{\boldsymbol{\theta}_{\mathcal{S}}}^T \overline{\mathbf{y}}\,\nabla_{\boldsymbol{\theta}_{\mathcal{S}}}\overline{\mathbf{y}} \succeq 0 \tag{26}$$

where $\overline{\mathbf{y}}_d$ is the stacked vector of desired model outputs, defined analogously to $\overline{\mathbf{y}}$. The matrix $\mathbf{H}(t)$ is the NTK at time $t$. The vector $\overline{\mathbf{y}}'$ follows a perturbed evolution

$$\dot{\overline{\mathbf{y}}}' = -\mathbf{H}'(t)\,(\overline{\mathbf{y}}' - \overline{\mathbf{y}}_d) + \overline{\mathbf{d}}(t) \ \text{ with } \ \mathbf{H}'(t) = \frac{1}{n}\nabla_{\theta'}^T \overline{\mathbf{y}}'\,\nabla_{\theta'}\overline{\mathbf{y}}' \succeq 0$$

with $\overline{\mathbf{d}}(t)$ is obtained by stacking the disturbance terms $\mathbf{d}_i(t)$, analogously to $\overline{\mathbf{y}}$ and $\overline{\mathbf{y}}_d$. Recent works have shown that for certain sufficiently overparameterized neural networks, $\mathbf{H}(t)$ remains strictly positive definite throughout the training process (Du et al., 2018; Arora et al., 2019; Huang & Yau, 2020; Liu et al., 2021). That is, there exists a strictly positive constant $\lambda_0 > 0$ such that

$$\mathbf{H}(t) \succeq \lambda_0 \tag{27}$$

Because the dynamics of $\overline{\mathbf{y}}$ are contracting with rate $\lambda_0$, and $\overline{\mathbf{y}} = \overline{\mathbf{y}}_d$ is a particular trajectory of the dynamics (since in that case $\frac{\mathrm{d}}{\mathrm{d}t}\overline{\mathbf{y}} = \mathbf{0}$), we can conclude that $\overline{\mathbf{y}}$ will converge towards $\overline{\mathbf{y}}_d$ exponentially. Similarly, $\overline{\mathbf{y}}'$ will converge to a ball of radius $D$ centered around $\overline{\mathbf{y}}_d$, where $||\overline{\mathbf{d}}|| \leq D$. We can determine $D$ as follows

$$||\overline{\mathbf{d}}|| = \sqrt{\sum_{j=1}^{n}||\mathbf{d}_i||^2} \leq \sqrt{\sum_{j=1}^{n}\frac{4\kappa^2 L^2}{n^2}} = \frac{2\kappa L}{\sqrt{n}} = D$$

where the first inequality was obtained by substituting in (25). This analysis shows that, after exponential transients of rate $\lambda_0$, we have that $\sup_i ||\mathbf{y}_i - \mathbf{y}_i'|| \leq ||\overline{\mathbf{y}} - \overline{\mathbf{y}}'|| \leq \frac{2\kappa L}{\lambda_0 \sqrt{n}}$. □

**Remark 9.** *The generalization bounds achieved above using output contraction scale as $1/\sqrt{n}$, while the generalization bounds achieved using parameter contraction (Theorem 4) scale as $1/n$. Intuitively, this is because the output vectors (21) interact with each other through equation (26). While resampling produces a disturbance of norm $1/n$ for each individual output vector, these disturbances can be amplified through coupling via (26). In particular, each individual disturbance term has a squared norm on the order of $1/n^2$. Adding these squared norms together, one finds that the total disturbance has a squared norm of the order of $n/n^2 = 1/n$. Thus, the total disturbance has a norm on the order of $1/\sqrt{n}$.*

**Equivalence of Polyak-Lojasiewicz and Output Contraction in the NTK Limit**  Here we relate the well-known Polyak-Lojasiewicz (PL) condition to contraction of the model outputs in the NTK setting. Specifically, we consider the notion of $\mu$-PL* introduced in (Liu et al., 2021). A loss landscape satisfies the $\mu$-PL* condition with $\mu > 0$ if

$$||\nabla\mathcal{L}||^2 \geq \mu\mathcal{L} \tag{28}$$

uniformly. It was shown in Liu et al. (2021), Theorem 1, that uniform positive definiteness of the Neural Tangent Kernel is sufficient for the mean-squared loss landscape to satisfy the $\mu$-PL* condition with $\mu = \lambda_0$, where $\lambda_0$ is a uniform lower bound on the smallest eigenvalue of Neural Tangent Kernel (27).

$$||\nabla_{\boldsymbol{\theta}}\mathcal{L}||^2 = (\overline{\mathbf{y}} - \overline{\mathbf{y}}_d)^T\mathbf{H}(t)(\overline{\mathbf{y}} - \overline{\mathbf{y}}_d) \geq \lambda_0||\overline{\mathbf{y}} - \overline{\mathbf{y}}_d||^2 = \lambda_0\mathcal{L} \tag{29}$$

Equation (29) shows that if the loss landscape $\mathcal{L}$ satisfies $\mu$-PL* uniformly, then the neural tangent kernel $\mathbf{H}(t)$ is uniformly positive definite, which in turn implies *output contraction*, i.e., contraction of the $\overline{\mathbf{y}}$ dynamics (26) in the section above. This is true for any nonlinear model, and does not require an infinite width assumption.

We now show in the wide network regime, where the Neural Tangent Kernel becomes constant during training, contraction of the model outputs is equivalent to the positive definiteness of the Neural Tangent Kernel. Thus, contraction of the model outputs implies the $\mu$-PL* condition. Indeed, the dynamics of the model outputs in the wide-network regime may be written as

$$\frac{\mathrm{d}}{\mathrm{d}t}(\overline{\mathbf{y}} - \overline{\mathbf{y}}_d) = -\mathbf{H}(\overline{\mathbf{y}} - \overline{\mathbf{y}}_d)$$

where $\mathbf{H} = \mathbf{H}^T$ is now constant. Because this is a linear time-invariant dynamical system, and $\mathbf{H}$ is symmetric, it is contracting if and only if $\mathbf{H}$ is positive definite – for linear systems, global asymptotic stability, global contraction, and eigenvalues with strictly negative real part are all equivalent conditions. Thus, in the wide network limit, where $\mathbf{H}$ is constant and symmetric, uniform positive definiteness of the Neural Tangent Kernel is equivalent to contraction of the model outputs.

**A Generalized Polyak-Lojasiewicz Condition with Metric**  The preceding paragraph establishes a connection between PL and output contraction. Given the prevalence of the metric in contraction analysis, this suggests generalizing the $\mu$-PL* condition to include explicit metric terms.

Assume for instance that the parameter vector $\boldsymbol{\theta}$ is being updated according to a *natural gradient* flow

$$\dot{\boldsymbol{\theta}} = -\mathbf{M}^{-1}(\boldsymbol{\theta})\nabla_{\boldsymbol{\theta}}\mathcal{L}$$

where $\mathbf{M}^{-1}(\boldsymbol{\theta}) \succeq \alpha\mathbf{I}$ is a symmetric positive definite matrix. The generalized PL condition

$$\nabla_{\boldsymbol{\theta}}\mathcal{L}^T\mathbf{Q}(\boldsymbol{\theta})\nabla_{\boldsymbol{\theta}}\mathcal{L} \geq \mu\mathcal{L}$$

where $0 < \mathbf{Q}(\boldsymbol{\theta}) \preceq \beta\mathbf{I}$ is bounded and symmetric, guarantees that the loss $\mathcal{L}$ converges exponentially to zero with rate $\alpha\mu/\beta$, as

$$\frac{\mathrm{d}}{\mathrm{d}t}\mathcal{L} = \nabla_{\boldsymbol{\theta}}\mathcal{L}^T\dot{\boldsymbol{\theta}} = -\nabla_{\boldsymbol{\theta}}\mathcal{L}^T\mathbf{M}(\boldsymbol{\theta})^{-1}\nabla_{\boldsymbol{\theta}}\mathcal{L} \leq -\frac{\alpha}{\beta}\nabla_{\boldsymbol{\theta}}\mathcal{L}^T\mathbf{Q}(\boldsymbol{\theta})\nabla_{\boldsymbol{\theta}}\mathcal{L} \leq -\frac{\alpha\mu}{\beta}\mathcal{L}$$

### 4.4  Partial Contraction and Virtual Systems

In some cases, a 'pure' contraction analysis may be hard to do. For example, when an optimizer has an adaptive learning rate, this can significantly complicate the calculation of the Jacobian. To deal with these difficulties, we make use of a generalization of contraction known as *partial contraction*, introduced in Wang & Slotine (2005) and further discussed in Jouffroy & Slotine (2004).

**Definition 3** (Partially Contracting Dynamical System)**.** Consider the system (3) (not necessarily contracting) and an auxiliary, virtual system of the form

$$\dot{\mathbf{y}} = \mathbf{g}(\mathbf{y}, \boldsymbol{\theta}, t)$$

Assume that this virtual system is contracting (with respect to $\mathbf{y}$) in metric $\mathbf{M}$ with rate $\lambda$, and also that $\mathbf{g}(\boldsymbol{\theta}, \boldsymbol{\theta}, t) = \mathbf{f}(\boldsymbol{\theta}, t)$. We then say that (3) is partially contracting. If one particular trajectory $\mathbf{y}(t)$ of the virtual system is known, then all trajectories of (3) converge exponentially towards $\mathbf{y}(t)$.

Robustness properties of a virtual contracting system lead directly to robustness properties for the original system (Del Vecchio & Slotine, 2012), which need not be contracting itself. This result provides a direct bridge for using virtual systems analysis to study generalization.

### 4.4.1 Adaptive Learning Rates

The results of the previous sections can be extended to include the presence of a state-dependent, time-varying, learning rate (Zeiler, 2012; Kingma & Ba, 2014; Goodfellow et al., 2016; Liu et al., 2019). In particular consider the learning dynamics with a learning rate scheduler $\rho(\boldsymbol{\theta}, t)$

$$\dot{\boldsymbol{\theta}}_S = -\rho(\boldsymbol{\theta}_S, t)\mathbf{G}(\boldsymbol{\theta}_S, S)$$

where $\rho_{max} \geq \rho(\boldsymbol{\theta}, t) \geq \rho_{min} > 0$. Now consider the *virtual system*

$$\dot{\boldsymbol{\theta}}_y = -\rho(\boldsymbol{\theta}_S, t)\mathbf{G}(\boldsymbol{\theta}_y, S)$$

If the $\boldsymbol{\theta}_y$ system is contracting in $\mathbf{M}$ with rate $\rho_{min}\lambda$, then we have, by the same arguments in Theorem 4, that $\mathcal{A}(S)$ is asymptotically (after exponential transients of rate $\rho_{min}\lambda$) uniformly $\epsilon$-stable with

$$\epsilon_{stab} = 2 \frac{\rho_{max}}{\rho_{min}} \frac{\chi L \xi}{\lambda n}$$

Note that the factor $\rho_{max}/\rho_{min} \geq 1$ plays a similar role to the condition number upper bound $\chi \geq 1$. It may be viewed as the "cost" of using a virtual system, similarly to that of switching to a new metric. This step is necessary because a naive application of contraction analysis on an optimizer with adaptive learning rates may be inconclusive. Using a virtual system provides a simple route toward obtaining the desired $1/n$ generalization bounds.

### 4.4.2 Sharpness Aware Minimization

Let us apply a simple partial contraction analysis to the recently introduced Sharpness Aware Minimization algorithm (SAM) (Foret et al., 2020). In continuous-time, SAM may be written

$$\dot{\boldsymbol{\theta}} = -\nabla_{\boldsymbol{\theta}}\mathcal{L}\big(\boldsymbol{\theta} + \rho\frac{\nabla_{\boldsymbol{\theta}}\mathcal{L}}{||\nabla_{\boldsymbol{\theta}}\mathcal{L}||}\big) \tag{30}$$

where $\rho > 0$. Assume first, slightly generalizing (Bartlett et al., 2022), that $\mathcal{L}$ is a convex quadratic loss of the form

$$\mathcal{L} = \frac{1}{2}\boldsymbol{\theta}^T\mathbf{P}(t)\boldsymbol{\theta}$$

with $\mathbf{P}(t)$ a symmetric uniformly positive definite matrix, i.e., $\mathbf{P}(t) \succeq p(t)\mathbf{I} \succeq p_{min}\mathbf{I} \succ \mathbf{0}$ where $p(t)$ and $p_{min}$ are scalars. Substituting in this loss function, the SAM optimizer dynamics may be written

$$\dot{\boldsymbol{\theta}} = -\mathbf{P}(t)\big(\boldsymbol{\theta} + \rho\frac{\mathbf{P}(t)\boldsymbol{\theta}}{||\mathbf{P}(t)\boldsymbol{\theta}||}\big) = -\big(\mathbf{P}(t) + \rho\frac{\mathbf{P}^2(t)}{||\mathbf{P}(t)\boldsymbol{\theta}||}\big)\boldsymbol{\theta} = -\mathbf{A}(\boldsymbol{\theta})\boldsymbol{\theta}$$

where $\mathbf{A}(\boldsymbol{\theta})$ is uniformly positive definite. It is readily shown that this optimizer is partially contracting in the identity metric, by using the virtual system

$$\dot{\boldsymbol{\theta}}_y = -\mathbf{A}(\boldsymbol{\theta})\boldsymbol{\theta}_y$$

which has $\boldsymbol{\theta}_y = \boldsymbol{\theta}$ and $\boldsymbol{\theta}_y = \mathbf{0}$ as particular solutions. Thus $\boldsymbol{\theta} \to \mathbf{0}$ exponentially with rate of at least $\lambda = p_{min} > 0$, which implies that SAM achieves a generalization rate on the order of $1/\lambda n$ for this loss function.

Consider now the case of an arbitrary (time-invariant) strongly convex loss, and modify the SAM dynamics (30) slightly for improved regularity to

$$\dot{\boldsymbol{\theta}} = -\nabla_{\boldsymbol{\theta}}\mathcal{L}\left(\boldsymbol{\theta} + \rho\frac{\nabla_{\boldsymbol{\theta}}\mathcal{L}}{||\nabla_{\boldsymbol{\theta}}\mathcal{L}|| + \epsilon}\right) \tag{31}$$

with $\epsilon > 0$ a small constant. Defining $\mathbf{e} = \frac{\nabla_{\boldsymbol{\theta}}\mathcal{L}}{\|\nabla_{\boldsymbol{\theta}}\mathcal{L}\|+\epsilon}$ , we then have

$$\nabla_{\boldsymbol{\theta}}\mathcal{L}(\boldsymbol{\theta} + \rho\mathbf{e}) = \nabla_{\boldsymbol{\theta}}\mathcal{L}(\boldsymbol{\theta}) + \int_0^\rho \nabla_{\boldsymbol{\theta}}^2\mathcal{L}(\boldsymbol{\theta} + s\mathbf{e})\,\mathbf{e}\,\mathrm{d}s = \nabla_{\boldsymbol{\theta}}\mathcal{L}(\boldsymbol{\theta}) + \rho\overline{\mathbf{H}}\mathbf{e}$$

where $\overline{\mathbf{H}}$ is the average Hessian over the line element, which is positive definite via strong convexity. This implies that $-\nabla_{\boldsymbol{\theta}}\mathcal{L}(\boldsymbol{\theta} + \rho\mathbf{e})$ is a descent direction for the loss,

$$-\nabla_{\boldsymbol{\theta}}\mathcal{L}(\boldsymbol{\theta})^T\nabla_{\boldsymbol{\theta}}\mathcal{L}(\boldsymbol{\theta} + \rho\mathbf{e}) = -\nabla_{\boldsymbol{\theta}}\mathcal{L}(\boldsymbol{\theta})^T\left(\mathbf{I} + \rho\frac{\overline{\mathbf{H}}}{\|\nabla_{\boldsymbol{\theta}}\mathcal{L}\|+\epsilon}\right)\nabla_{\boldsymbol{\theta}}\mathcal{L}(\boldsymbol{\theta}) \leq -\|\nabla_{\boldsymbol{\theta}}\mathcal{L}(\boldsymbol{\theta})\|^2$$

This in turn implies that the modified SAM dynamics (31) converges to the unknown global optimizer $\boldsymbol{\theta}^*$. Similarly to the previous example, consider now the virtual system

$$\dot{\boldsymbol{\theta}}_y = -\mathbf{A}(\boldsymbol{\theta})\nabla_{\boldsymbol{\theta}}\mathcal{L}(\boldsymbol{\theta}_y) \quad\text{with}\quad \mathbf{A}(\boldsymbol{\theta}) = \mathbf{I} + \rho\frac{\overline{\mathbf{H}}(\boldsymbol{\theta})}{\|\nabla_{\boldsymbol{\theta}}\mathcal{L}(\boldsymbol{\theta})\|+\epsilon} \tag{32}$$

This virtual system has $\boldsymbol{\theta}_y = \boldsymbol{\theta}$ and $\boldsymbol{\theta}_y = \boldsymbol{\theta}^*$ as particular solutions. Furthermore, considering for this system the metric $\mathbf{M}(\boldsymbol{\theta}) = \mathbf{A}^{-1}(\boldsymbol{\theta})$ (which is purely time-dependent to the virtual system), and denoting $\mathbf{f}_y = \dot{\boldsymbol{\theta}}_y$ ,

$$\mathbf{M}(\boldsymbol{\theta})\frac{\partial\mathbf{f}_y}{\partial\boldsymbol{\theta}_y}(\boldsymbol{\theta}_y) + \frac{\partial\mathbf{f}_y}{\partial\boldsymbol{\theta}_y}(\boldsymbol{\theta}_y)^T\mathbf{M}(\boldsymbol{\theta}) + \dot{\mathbf{M}}(\boldsymbol{\theta}) = -2\nabla^2\mathcal{L}(\boldsymbol{\theta}_y) - \mathbf{A}^{-1}(\boldsymbol{\theta})\dot{\mathbf{A}}(\boldsymbol{\theta})\mathbf{A}^{-1}(\boldsymbol{\theta})$$

Letting $\beta = \lambda_{min}[\nabla^2\mathcal{L}(\boldsymbol{\theta}_y)]\,\lambda_{min}[\mathbf{A}(\boldsymbol{\theta})]$ , the above implies [1]

$$\mathbf{M}(\boldsymbol{\theta})\frac{\partial\mathbf{f}_y}{\partial\boldsymbol{\theta}_y}(\boldsymbol{\theta}_y) + \frac{\partial\mathbf{f}_y}{\partial\boldsymbol{\theta}_y}(\boldsymbol{\theta}_y)^T\mathbf{M}(\boldsymbol{\theta}) + \dot{\mathbf{M}}(\boldsymbol{\theta}) \leq -2\beta\mathbf{M}(\boldsymbol{\theta}) - \mathbf{A}^{-1}(\boldsymbol{\theta})\dot{\mathbf{A}}(\boldsymbol{\theta})\mathbf{A}^{-1}(\boldsymbol{\theta})$$

As $\boldsymbol{\theta}(t)$ approaches the global optimum $\boldsymbol{\theta}^*$, $\overline{\mathbf{H}}(\boldsymbol{\theta})$ reaches some limit and $\|\nabla_{\boldsymbol{\theta}}\mathcal{L}(\boldsymbol{\theta})\|$ goes to zero, so that $\dot{\mathbf{A}}(\boldsymbol{\theta})$ goes to zero. Let $h_{min}^*$ and $h_{max}^*$ be the minimum and maximum eigenvalue of the Hessian $\nabla_{\boldsymbol{\theta}}^2\mathcal{L}$ at $\boldsymbol{\theta}^*$. The virtual system becomes contracting with rate $h_{min}^*(1 + \rho\,h_{min}^*/\epsilon)$ , so that including the conditioning term $\chi$ and the disturbance term $D \leq 2\|\mathbf{A}(\boldsymbol{\theta})\|L$ yields by Theorem (2) the generalization bound

$$\frac{2\chi(1 + \rho h_{max}^*/\epsilon)L^2}{\lambda_\infty n} = \sqrt{\frac{1 + \rho\frac{h_{max}^*}{\epsilon}}{1 + \rho\frac{h_{min}^*}{\epsilon}}}\,\frac{2(1 + \rho h_{max}^*/\epsilon)L^2}{h_{min}^*(1 + \rho\,h_{min}^*/\epsilon)n} \approx \left(\frac{h_{max}^*}{h_{min}^*}\right)^{3/2}\frac{2L^2}{h_{min}^* n}$$

where we have approximated the expression for small $\epsilon$. Aside from a constant prefactor of $\left(\frac{h_{max}^*}{h_{min}^*}\right)^{3/2}$, this is the same generalization bound and desirable $1/n$ scaling as in standard gradient descent. Note that this result, as far as we aware, provides the first global convergence analysis and generalization bound for SAM in a distribution-free, general convex setting. The empirical observation that SAM generalizes better than gradient descent (Foret et al., 2020) suggests that this bound may be further tightened.

## 5 Concluding Remarks

**Is Contraction the Correct Approach to Studying Generalization?** In deep learning, models are trained using gradient descent on non-convex loss functions. Despite the non-convexity of the loss, these models achieve low test error. However, most theoretical analyses of generalization are only applied to convex loss landscapes. This disconnect suggests the need for more general nonlinear analysis tools which can provide tighter generalization bounds for non-convex landscapes (Foret et al., 2020; Bartlett et al., 2022). We demonstrate that contraction analysis is one such tool, although of course it may only provide part of the picture. Contraction analysis can be applied either directly to optimizer dynamics (as in Theorem 4), or to the outputs of a model being trained with gradient flow (as in the Neural Tangent Kernel examples). In both

---

[1]Since both $\nabla^2\mathcal{L}$ and $\mathbf{A}$ are symmetric positive definite, $\nabla^2\mathcal{L}(\boldsymbol{\theta}_y) \geq \lambda_{min}(\nabla^2\mathcal{L}(\boldsymbol{\theta}_y))\mathbf{I} = \beta\,1/\lambda_{min}(\mathbf{A}(\boldsymbol{\theta}))\mathbf{I} \geq \beta\mathbf{A}^{-1}(\boldsymbol{\theta})$

cases, contraction yields generalization certificates that improve as the number of training samples increases. Although overparameterized optimizers cannot be globally contracting (because overparameterization implies many possible optima), it is expected that the outputs of an overparameterized model will be contracting as the loss converges to zero. Contraction is also consistent with training curves that do not look like pure exponential decay. This is because contraction analysis yields *upper bounds* on the contraction rate; training curves are free to be arbitrarily complicated as long as they are upper-bounded by an exponential decay function.

**Why Metrics are Critical**   Metrics play a crucial role in our analysis. It is natural to wonder whether changing the metric (or contraction rate) could improve optimization but harm generalization (Chen et al., 2018). However, since metrics are an analysis tool (i.e., they do not influence the optimization process, only how it is analyzed), their use in analyzing generalization has no bearing on the performance of the optimizer. That said, as section 3.2 shows, a "bad" choice of metric can significantly alter the stability and generalization bounds for a given optimizer. Indeed, the dependence of stability measurement on coordinate systems is precisely why metrics are necessary. Contraction analysis considers differential coordinate transforms, which provide a more flexible set of tools than other stability analyses that only consider explicit coordinate transforms (Lohmiller & Slotine, 1998).

**Comparison to Related Work**   Our results are similar in spirit to Charles & Papailiopoulos (2018), in the sense that we also use an optimizer's intrinsic dynamical stability to provide generalization error bounds. In certain cases–for example gradient flow on strongly convex losses–our results allow us to derive tighter bounds, because we do not assume the existence of a global minimizer $\boldsymbol{\theta}^*$ and go through the triangle inequality to bound the distance between $\boldsymbol{\theta}_S$ and $\boldsymbol{\theta}_{S'}$.

**Future Directions**   Contracting systems are robust to noise (Pham & Slotine, 2013), and therefore it seems likely that the results presented here can straightfowardly be extended to stochastic gradient flows– along the lines of Mandt et al. (2015) or Boffi & Slotine (2020). This may be applicable to settings where gradient training may be seen as Wasserstein gradient flow (Bouvrie & Slotine, 2019; Mei et al., 2019; Chizat, 2022). Our work also suggests a potential connection to the double descent phenomenon (Nakkiran et al., 2021). In particular (14) implies that the generalization error can overshoot by a factor of $L\chi$, which gives room for the generalization to increase transiently from its initial value before it eventually decreases. It has recently been shown that gradient training of deep linear networks is related to Riemannian gradient flow (Cohen et al., 2022), another potential application of our results relating contraction to generalization. Similarly, it has been shown in Bernacchia et al. (2018) that training deep linear networks with natural gradient descent leads to contraction (in the identity metric) of the network weights towards their optimal value. Theorem 4 is immediately applicable to this setting. Additionally, it has been shown that the self-attention mechanism of Transformers may be interpreted as a primal-dual algorithm (Nguyen et al., 2023). Given the correspondence between primal-dual algorithms and contraction (Nguyen et al., 2018), and in particular between pre-conditioned primal-dual algorithms and Riemannian contraction (Wensing & Slotine, 2020, Section 3.2), it could be interesting to also analyze Transformers through a contraction lens.

Finally, recent extensions of Riemannian contraction to general Banach spaces (Srinivasan & Slotine, 2023) may allow further insights in this broad context.

We conclude with some speculations on the potential connection between our results and biology, specifically neuroscience. The role of non-Euclidean geometry in the objective-based functions of the brain remains an open question (Surace et al., 2020). Many local synaptic rules can be viewed as implementing optimization over a loss function, such as Hebbian plasticity minimizing Principal Component Loss in certain settings (Oja, 1992). It seems plausible that our results can be used to quantify the generalization behavior of such rules. Additionally, we did not explore the combination properties of contracting systems, which can be combined in various hierarchical and feedback forms that automatically preserve contraction (Lohmiller & Slotine, 1998; Slotine & Lohmiller, 2001; Kozachkov et al., 2021). Our results suggest that combinations of contracting optimizers automatically generalize well, a property that evolution would likely preserve in a system like the brain.

## 6    Acknowledgments

This work benefited from stimulating discussions with John Tauber (Neuroscience Statistics Lab at MIT), Akshay Rangaman, Andrzej Banburski-Fahey (Center for Brains, Minds and Machines at MIT), as well as members of the Fiete Lab at MIT. We also thank reviewers for suggestions that strengthened the paper.

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

# A  Appendix

## A.1  Proof of (8)

*Proof.* For every time $t$ we randomly sample $b$ indices $i_1, \cdots, i_b$. Using these indices we select datapoints $z_i, \cdots, z_{i_b}$ from $S$ and $S'$ to update $\boldsymbol{\theta}_t^S$ and $\boldsymbol{\theta}_t^{S'}$ respectively. At every time $t$ there are two possibilities. Either we do not draw the replaced element $z_k'$ or we do. Denote these events $A$ and $B$, respectively (Figure

5). We have $P(A) = 1 - \frac{b}{n}$ and $P(B) = \frac{b}{n}$. If event $A$ occurs, then by assumption we expect the geodesic distance to shrink

$$\mathbb{E}_{\mathcal{A}}[d_{\mathcal{M}_{t+1}}(\boldsymbol{\theta}_{t+1}^S, \boldsymbol{\theta}_{t+1}^{S'})|A] \leq \mu \mathbb{E}_{\mathcal{A}}[d_{\mathcal{M}_t}(\boldsymbol{\theta}_t^S, \boldsymbol{\theta}_t^{S'})|A] \tag{33}$$

where $\mathbb{E}[\cdot|A]$ denotes the conditional expectation given event $A$. However, if the replaced element is drawn (i.e., event $B$ occurs) then we have

$$\boldsymbol{\theta}_{t+1}^S = \frac{1}{b}\sum_{i=1}^{b} \mathbf{g}(\boldsymbol{\theta}_t^S, z_i) \equiv \hat{\mathbf{G}}(\boldsymbol{\theta}_t^S)$$

$$\boldsymbol{\theta}_{t+1}^{S'} = \hat{\mathbf{G}}(\boldsymbol{\theta}_t^{S'}) + \mathbf{d}(\boldsymbol{\theta}_t^{S'})$$

where $\mathbf{d}(\boldsymbol{\theta}_t^{S'}) = \frac{1}{b}(\mathbf{g}(\boldsymbol{\theta}_t^{S'}, z_k') - \mathbf{g}(\boldsymbol{\theta}_t^{S'}, z_k))$. As in Theorem (4), we have written the update for $\boldsymbol{\theta}^{S'}$ as a 'perturbed' version of the update for $\boldsymbol{\theta}^S$. We will now derive an analogous robustness result, and then use the linearity of expectation to bound the overall geodesic distance. Note that

$$\begin{aligned}
d_{\mathcal{M}_{t+1}}(\boldsymbol{\theta}_{t+1}^S, \boldsymbol{\theta}_{t+1}^{S'}) &= d_{\mathcal{M}_{t+1}}(\hat{\mathbf{G}}(\boldsymbol{\theta}_t^S), \hat{\mathbf{G}}(\boldsymbol{\theta}_t^{S'}) + \mathbf{d}(\boldsymbol{\theta}_t^{S'})) \\
&\leq d_{\mathcal{M}_{t+1}}(\hat{\mathbf{G}}(\boldsymbol{\theta}_t^S), \hat{\mathbf{G}}(\boldsymbol{\theta}_t^{S'})) + d_{\mathcal{M}_{t+1}}(\hat{\mathbf{G}}(\boldsymbol{\theta}_t^{S'}), \hat{\mathbf{G}}(\boldsymbol{\theta}_t^{S'}) + \mathbf{d}(\boldsymbol{\theta}_t^{S'})) \\
&\leq d_{\mathcal{M}_{t+1}}(\hat{\mathbf{G}}(\boldsymbol{\theta}_t^S), \hat{\mathbf{G}}(\boldsymbol{\theta}_t^{S'})) + \sqrt{M_{max}}\frac{2\xi}{b}
\end{aligned}$$

where the first inequality comes from the triangle inequality and the second comes from the boundedness of $\mathbf{d}(\boldsymbol{\theta}_t^{S'})$ and the metric distortion bound in Theorem 3. Now applying the assumption of contraction in expectation we get

$$\begin{aligned}
\mathbb{E}_{\mathcal{A}}[d_{\mathcal{M}_{t+1}}(\boldsymbol{\theta}_{t+1}^S, \boldsymbol{\theta}_{t+1}^{S'})|B] &\leq \mathbb{E}_{\mathcal{A}}[d_{\mathcal{M}_{t+1}}(\hat{\mathbf{G}}(\boldsymbol{\theta}_t^S), \hat{\mathbf{G}}(\boldsymbol{\theta}_t^{S'}))|B] + \sqrt{M_{max}}\frac{2\xi}{b} \\
&\leq \mu \mathbb{E}_{\mathcal{A}}[d_{\mathcal{M}_t}(\boldsymbol{\theta}_t^S, \boldsymbol{\theta}_t^{S'})|B] + \sqrt{M_{max}}\frac{2\xi}{b}
\end{aligned} \tag{34}$$

We can now use the linearity of the expectation operator to bound the geodesic distance, and then use the metric distortion result to bound the Euclidean distance

$$\begin{aligned}
\mathbb{E}_{\mathcal{A}}[d_{\mathcal{M}_{t+1}}(\boldsymbol{\theta}_{t+1}^S, \boldsymbol{\theta}_{t+1}^{S'})] &= \mathbb{E}_{\mathcal{A}}[d_{\mathcal{M}_{t+1}}(\boldsymbol{\theta}_{t+1}^S, \boldsymbol{\theta}_{t+1}^{S'})|A]P(A) + \mathbb{E}_{\mathcal{A}}[d_{\mathcal{M}_{t+1}}(\boldsymbol{\theta}_{t+1}^S, \boldsymbol{\theta}_{t+1}^{S'})|B]P(B) \\
&\leq \mu \mathbb{E}_{\mathcal{A}}[d_{\mathcal{M}_t}(\boldsymbol{\theta}_t^S, \boldsymbol{\theta}_t^{S'})](1 - \frac{b}{n}) + (\mu \mathbb{E}_{\mathcal{A}}[d_{\mathcal{M}_t}(\boldsymbol{\theta}_t^S, \boldsymbol{\theta}_t^{S'})] + \sqrt{M_{max}}\frac{2\xi}{b})\frac{b}{n} \\
&= \mu \mathbb{E}_{\mathcal{A}}[d_{\mathcal{M}_t}(\boldsymbol{\theta}_t^S, \boldsymbol{\theta}_t^{S'})] + \sqrt{M_{max}}\frac{2\xi}{n}
\end{aligned}$$

Using the metric distortion bounds and unraveling the recursion yields

$$\mathbb{E}_{\mathcal{A}}[d(\boldsymbol{\theta}_t^S, \boldsymbol{\theta}_t^{S'})] \leq \chi \mu^t C + \frac{2\chi\xi}{(1-\mu)n}$$

Where $\chi = \sqrt{\frac{M_{max}}{M_{min}}}$ has again come from the metric distortion bound. Multiplying through by $L$, we have that

$$\epsilon_{stab} = \frac{2L\chi\xi}{n(1-\mu)}$$

which is the same result as the continuous-time case, except that $\lambda \rightarrow (1-\mu)$. $\qquad\square$

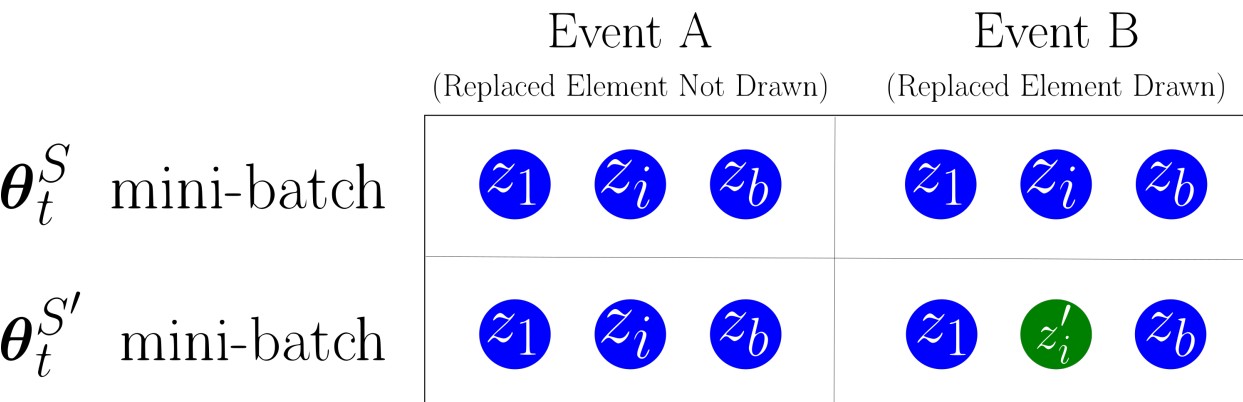

Figure 5: Illustration of the two cases for updating with mini-batches. At each time $t$, we randomly sample $b$ indices between 1 and $n$. Then we draw the corresponding datapoints from sets $S$ and $S'$ to form the mini-batches used to update $\boldsymbol{\theta}_t^S$ and $\boldsymbol{\theta}_t^{S'}$ respectively. In Event A (left column), the index of the replaced element is not selected, and therefore the datapoints used to update $\boldsymbol{\theta}_t^S$ and $\boldsymbol{\theta}_t^{S'}$ are the same. In Event B, the index of the replaced element is selected, and so the datapoints used to perform the update are different.

