# OpenReview forum: "Generalization as Dynamical Robustness--The Role of Riemannian Contraction in Supervised Learning"
_TMLR — Accepted by TMLR_

### Review · Reviewer_7zJi · 2022-12-03

**Summary Of Contributions:**

The paper extended the framework of Hardt et al. [1] to the Riemannian manifold setting. In the Euclidean setting, this framework provides a generalization bound via uniform stability. In particular, if the loss is strongly convex and has Lipschitz gradients, one can achieve a bound only dependent on the condition number of the loss and number of samples. The analysis of stability is essentially due to contraction - the algorithm cannot change too much to begin with, and therefore it cannot change much more after swapping out a single data point. The authors then proceeds to analyze several examples that fits into this framework but do not fit into the Euclidean setting.

References
1. Hardt, Moritz, Ben Recht, and Yoram Singer. "Train faster, generalize better: Stability of stochastic gradient descent." In International conference on machine learning, pp. 1225-1234. PMLR, 2016.


**Audience:**

Yes

**Claims And Evidence:**

Yes

**Requested Changes:**

Aesthetically, there is definitely a question of whether or not this is the "correct approach" to studying generalization. While I believe this is of a lesser concern for TMLR, I believe the authors should at least add a discussion towards this direction for readers who are unfamiliar with the meta concerns of this approach.

Is contraction really the reason why gradient descent type algorithms generalize? The main criticism of Hardt et al.'s approach is that in practice we often run gradient based algorithms in nonconvex settings, which do not yield contractions in the first place. This is in sharp contrast with the empirical observations that generalization does still happen without contraction - at least the training curves typically do not look like exponential decay. I hope the authors can provide some discussion, not necessarily a rebuttal, but at least to help put this paper's contribution in appropriate context.

At the same time, I was hoping the authors consider some other interesting examples of manifolds in their paper, which I believe will significantly strengthen the contributions:
1. Neural tangent kernel training is contracting on a Hilbert space, a special case of Riemannian manifold, and the contraction rate depends on the minimum eigenvalue of the Gram matrix.
2. Gradient training of deep linear networks is Riemannian gradient flow, the contraction depends on the strong convexity of the loss function e.g. least squares [2].
3. Gradient training of two layer mean field networks is Wasserstein gradient flow [3], although contraction is an open problem, see [4] for some progress towards this direction.

I would strongly recommend adding at least some discussion towards these connections and ideally the first two examples.

References
2. Cohen, Nadav, Govind Menon, and Zsolt Veraszto. "Deep Linear Networks for Matrix Completion--An Infinite Depth Limit." arXiv preprint arXiv:2210.12497 (2022).
3. Mei, Song, Theodor Misiakiewicz, and Andrea Montanari. "Mean-field theory of two-layers neural networks: dimension-free bounds and kernel limit." In Conference on Learning Theory, pp. 2388-2464. PMLR, 2019.
4. Chizat, Lénaïc. "Mean-field langevin dynamics: Exponential convergence and annealing." arXiv preprint arXiv:2202.01009 (2022).

**Strengths And Weaknesses:**

Strengths

The results are correct and easily verifiable. The idea of the paper is somewhat obvious in hindsight, but I believe the consequences are significant as it significantly improves the limitations of the Hardt et al. results in the strong convex loss setting.

Weaknesses

Intuitively speaking, the extension to the Riemannian manifold setting is unsurprising - contraction on any metric space will yield uniform stability bounds. Therefore I personally do not believe this is a particularly strong technical contribution.

---

> ### Author Response · Authors · 2022-12-15
> **Clarifying Question Regarding Point (1)**
>
> We thank the reviewer for their thoughtful suggestions and references. We have a clarifying question regarding point (1), on the topic of Neural Tangent Kernel training. This sounds very interesting, but we are unaware of this result. Would the reviewer be able to provide a representative reference?

---

> > ### Comment · Reviewer_7zJi · 2022-12-16
> > **NTK Manifold**
> >
> > I don't know of any papers that explicitly made this idea precise, however I'm sure there are works in kernel methods that have noticed this structure, and many experts know this as a "folklore". One paper that did mention this at a high level is in Hanin and Nica [1], see the point (ii) at the top of page 2 for the comment.
> >
> > That being said, it's fairly easy to give a precise description of this manifold structure - which holds for all kernel methods - and I will describe this below. I personally like the notation and description from Huang and Yau [2], so the comments below will be somewhat based on this.
> >
> > (I also apologize the unnecessary line breaks, as OpenReview's Markdown system is kind of buggy when it comes to the math, and this was the only quick way I found to fix it)
> >
> > If we let $(x^{\alpha}, y^{\alpha} )_{\alpha=1}^m$
> >
> > denote the input/output data points, and we denote $\{f^\alpha_t = f(x^\alpha, \theta_t) \}_{\alpha=1}^m$
> >
> > denote the neural network output at time $t\geq 0$, where time $t$ denotes training under gradient flow $\partial_t \theta_t = - \nabla L(\theta_t)$ for least square loss $L$. In this notation, we have that (see equation 1.8 of [2])
> >
> > $\partial_t (f^\alpha_t - y^\alpha) = - \sum_{\beta=1}^m K^{(2)}_t( x^\alpha, x^\beta ) ( f^\beta_t - y^\beta ) $
> >
> > where $ K^{(2)}_t( x^\alpha, x^\beta ) = \langle \nabla_\theta f^\alpha_t , \nabla_\theta f^\beta_t \rangle $ is the neural tangent kernel.
> >
> > The main observation is that in the infinite width limit, $K^{(2)}_t$
> >
> > is deterministic and constant in time, which is a natural candidate for the inverse Riemannian metric $g^{\alpha\beta}$. More precisely, if our coordinates are $( f^\alpha - y^\alpha )_{\alpha=1}^m$, then the above equation is exactly **Riemannian gradient flow** with respect to the inverse metric $K^{(2)}_t$.
> >
> > Furthermore, we also observe that since the metric does not depend on the coordinates $( f^\alpha - y^\alpha )_{\alpha=1}^m$,
> >
> > then this merely defines a Hilbert space, instead of a manifold. A trivial way to check is the Christoffel symbols all vanish since $\partial_\gamma g_{\alpha\beta} = 0$, therefore this manifold is flat.
> >
> > 1. https://arxiv.org/abs/1909.05989
> >
> > 2. https://arxiv.org/abs/1909.08156

---

> > > ### Author Response · Authors · 2023-03-06
> > > **Thank you for the wonderful suggestions and comments, as well as the helpful follow-up reply. We respond below:**
> > >
> > > | Is contraction really the reason why gradient descent type algorithms generalize? The main criticism of Hardt et al.'s approach is that in practice we often run gradient based algorithms in nonconvex settings, which do not yield contractions in the first place. This is in sharp contrast with the empirical observations that generalization does still happen without contraction - at least the training curves typically do not look like exponential decay. I hope the authors can provide some discussion, not necessarily a rebuttal, but at least to help put this paper's contribution in appropriate context.
> > >
> > > Thank you for this very interesting question and comment. We agree that a discussion of these points would improve the paper, and have included them in the Discussion section, in the paragraph entitled “Is Contraction the Correct Approach to Study Generalization?”
> > >
> > > | At the same time, I was hoping the authors consider some other interesting examples of manifolds in their paper…
> > >
> > > Thank you for these wonderful suggested applications. We decided to focus on Neural Tangent Kernel, your first suggestion, because it opened the most doors to new results. We appreciate the pointers from the reviewer that helped us to see this connection more clearly. We have added a new section “Output Contractions, Neural Tangent Kernels, and Polyak-Lojasiewicz”. We have added your other two suggestions in the “Future Work” section of the discussion.

---

> > > > ### Comment · Reviewer_7zJi · 2023-03-06
> > > > **Response**
> > > >
> > > > Thank you for address my comments. I'm happy with the changes and would recommend accept of this paper.
> > > >
> > > > Just a minor comment, the figures are not showing up correctly in the current pdf file. You may want to update that soon.

---

> > > > > ### Author Response · Authors · 2023-03-06
> > > > > **Thank you!**
> > > > >
> > > > > Thanks very much for pointing this out, the problem has been fixed.
> > > > >
> > > > >
> > > > > Also thank you for recommending acceptance and (again) thank you for your excellent feedback & suggestions.

---

### Review · Reviewer_8Jkf · 2022-12-06

**Summary Of Contributions:**

This paper proves the algorithmic stability of certain flow-like optimization algorithms (in particular SGD and Natural Gradient). This is done by rewriting the (Euclidean) problem in the lens of a dynamical system with a new Riemannian metric and showing contraction under this new metric. Results are shown on a variety of learning problems. Furthermore, the paper then expands the notion of "contractiveness" to be more inclusive, allowing for some analysis on other optimization problems.

**Audience:**

Yes

**Claims And Evidence:**

Yes

**Requested Changes:**

Mostly, could you address my questions in "weaknesses".

**Strengths And Weaknesses:**

Strengths
-----------
* The paper analyzes a rather classical question (algorithmic generalization) with a somewhat unique perspective. The insights are reasonably deep and provide a good glimpse into several common optimization problems.
* The derivations are concise, easy to follow, and are (at least to my current calculation) correct.
* The results seem reasonably strong. In particular, the remarks on page 5 and 6 help contextualize the work. In particular, the derived bounds provide good insight into general algorithmic stability open problems.
* The use-cases showcase a broad applicability to several classical problems.

Weaknesses
--------------
* I found it rather hard to follow the notation. One notable example is that "x" and "$\theta$" are both used to describe points on the underlying space, a pretty hard to follow overloading. I would recommend the authors generally revise to increase clarity.
* In Theorem 2, the condition on the metrics imply that the metrics do not vanish or explode at infinity. I wonder how realistic this is; does it hold, for example, for the metrics defined in the experiments?
* A similar question should be asked for Theorem 4. In this case, one should also be careful with how the metrics evolve with time, so a comment about it here would also be nice.

---

> ### Author Response · Authors · 2023-03-06
> **Thank you for the helpful comments and suggestions, we respond below:**
>
> | I found it rather hard to follow the notation…
>
> Thank you for pointing this out. We agree, and have corrected the text accordingly to simply use $\theta$ everywhere.
>
> | In Theorem 2, the condition on the metrics imply that the metrics do not vanish or explode at infinity. I wonder how realistic this is; does it hold, for example, for the metrics defined in the experiments? A similar question should be asked for Theorem 4. In this case, one should also be careful with how the metrics evolve with time, so a comment about it here would also be nice.
>
> Thank you for the interesting and helpful comment. To define a proper metric, the matrix $\mathbf{M}(\mathbf{\theta},t)$ must be uniformly positive definite for all $\mathbf{\theta}$ and $t$. This ensures that the metric cannot vanish at infinity.
>
> We can expect the metric to not explode at infinity when, for example, the norm of the metric is Lipshitz with respect to the system state $\mathbf{\theta}$:
>
> $ \[||\mathbf{M}(\mathbf{\theta},t)|| \leq k ||(\mathbf{\theta}|| \]$
>
> As discussed in the text, a contracting optimizer will remain in a finite set, ensuring that the matrix norm above stays bounded. As a concrete example, the metric used for the Rosenbrock function in the section “The Choice of Metric is Critical” satisfies this property.
>
> We have added this as a comment into the section on Geodesics and Bounded Distortions.

---

### Review · Reviewer_yqhG · 2023-02-20

**Summary Of Contributions:**

The paper studies generalization guarantees for optimization algorithms based on algorithmic stability, with a focus on algorithms that contract trajectories in some Riemannian metric. After providing background on Riemannian contractions, the authors present new algorithmic stability bounds for deterministic and stochastic contracting optimizers, which leverage such Riemannian metrics. They then provide several examples to illustrate the theory, as well as extensions to settings where contractions may only hold in a weaker sense.

**Audience:**

Yes

**Claims And Evidence:**

Yes

**Requested Changes:**

The main change I'd like to request is described above, i.e. finding examples where the proposed bounds can actually improve over trivial metrics. This seems critical.

Another important aspect to discuss is regarding the trade-offs between stability and optimization, following related works such as Chen et al, 2018: are there cases where changing the metric or contraction rate may improve optimization but hurt stability, or vice versa, and how may such trade-offs be addressed in this context?

Some additional minor comments/typos:
- "Definition 1.1": the numbering seems different between Definitions and other results (e.g. Theorem 1)
- Remark 3: dynamics *in* equation..
- Theorem 4: please add comments: how does this bound compare to the full-batch case? how does the mini-batch size affect the involved quantities, such as $\mu$?
- Remark 5: do you have any specific examples where improved stability bounds can be obtained by leveraging the varying metric?
- section 4.2.1: "and setting $R(0) = 0$"
- section 4.3.1: again the stability bound seems worse when the learning rate is non-contant (due to $\rho_{max} / \rho_{\min} \geq 1$), which suggests the proposed theory does not explain any benefits of adaptive learning rates, so perhaps this section is not very useful here?

**Strengths And Weaknesses:**

Leveraging different metrics for better generalization guarantees is a useful direction in the context of algorithmic stability, and may help design better algorithms. The present work provides a new perspective on this based on Riemannian contractions, and is thus of interest for the community. The paper is also well written, and the presentation is clear.

Nevertheless, the current manuscript does not highlight cases where using a non-trivial Riemannian metric is actually helpful for improving generalization bounds, which seems like an important weakness. In particular, the examples in Section 3.1 and 3.2 seem to suggest that the best bounds are achieved when using the trivial metric (M = I) or preconditioner (P = I). It seems important to discuss an example where a non-trivial metric may help with generalization, perhaps with additional assumptions (see, e.g., [this paper](https://arxiv.org/abs/2006.10732) for an example where preconditioning may help, though with different analyses). If this is difficult in the current framework, it would be great to identify why.

---

> ### Author Response · Authors · 2023-03-06
> **Thank you for the helpful suggestions and comments, we respond below:**
>
> | the current manuscript does not highlight cases where using a non-trivial Riemannian metric is actually helpful for improving generalization bounds, which seems like an important weakness…The main change I'd like to request is described above, i.e. finding examples where the proposed bounds can actually improve over trivial metrics.
>
> We thank the reviewer for their helpful suggestion. We agree that this would improve the quality and presentation of the paper. We have added an analytical example of when picking the “wrong” metrics leads to inconclusive stability and generalization bounds, and picking the “right” metric leads to the usual 1/n generalization bounds.  Please see the section “The Choice of Metric is Critical”.
>
> | Another important aspect to discuss is regarding the trade-offs between stability and optimization, following related works such as Chen et al, 2018: are there cases where changing the metric or contraction rate may improve optimization but hurt stability, or vice versa, and how may such trade-offs be addressed in this context?
>
> We thank the reviewer for pointing this out. We have added a new paragraph into the discussion highlighting the fact that metrics are analysis tools, and thus cannot hurt or improve optimizers in any intrinsic sense. Metrics can only harm or improve the stability and generalization bounds one obtains for a given optimizer.
>
> Theorem 3 in our paper shows that, aside from constant prefactors (i.e., $\lambda$ and $\chi$), choosing a different metric that the optimizer is contracting in will not alter the 1/n rate of the generalization bound.
>
> | "Definition 1.1": the numbering seems different between Definitions and other results (e.g. Theorem 1)
>
> Thanks, fixed.
>
> | Remark 3: dynamics in equation..
>
> Fixed.
>
> | Theorem 4: please add comments: how does this bound compare to the full-batch case? how does the mini-batch size affect the involved quantities…
>
> We have added a remark after Theorem 4 explaining why the batch size, $b$, does not appear in the generalization bound.
>
> | Remark 5: do you have any specific examples where improved stability bounds can be obtained by leveraging the varying metric?
>
> Yes, please see the new section “The Choice of Metric is Critical”.
>
> | section 4.3.1: again the stability bound seems worse when the learning rate is non-contant…
>
> Thank you for pointing this out. The main added value of this section is being able to prove the optimizer is contracting at all. This is similar to the discussion about metrics above. In this case, a naive application of contraction analysis (without a virtual system) might lead to inconclusive stability and generalization results. We have added a comment into the section to better highlight this.

---

### Decision · Action_Editors · 2023-04-04

**Recommendation:** Accept as is

**Comment:**

This paper studies generalisation bounds for supervised learning based on the algorithmic stability framework of Bousquet & Elisef , using contraction properties of the algorithm (e.g a gradient flow) in appropriate riemanninan metrics.

Reviewers  agreed that this is submission puts forward an interesting and clear extension of the notion of algorithmic stability. Despite the relative lack of novel technical contributions, overall they found this to be a valuable contribution.

**Audience:**

Yes, this is appealing to a general audience interested in ML foundations.

**Claims And Evidence:**

Yes, they are supported by mathematical statements, and numerous examples are provided.

---

> ### Author Response · Authors · 2023-04-10
> **Submitted Camera-Ready Version + thank you**
>
> Dear AEs and Reviewers,
>
> We thank you for accepting our paper, as well as for the high quality of reviews we received.
>
> We have submitted the camera-ready version of our manuscript.
>
> - The Authors